# Flexible graphene-based neurotechnology for high-precision deep brain mapping and neuromodulation in Parkinsonian rats

Nicola Ria [1], Ahmed Eladly [2], Eduard Masvidal-Codina[1], Xavi Illa [3,4], Anton Guimerà[3,4], Kate Hills [2], Ramon Garcia-Cortadella [1,5], Fikret Taygun Duvan [1], Samuel M. Flaherty[2], Michal Prokop[1], Rob. C. Wykes[2,6] ✉, Kostas Kostarelos[1,2,7,8] ✉ & Jose A. Garrido [1,7] ✉

Deep brain stimulation (DBS) is a neuroelectronic therapy for the treatment of a broad range of neurological disorders, including Parkinson's disease. Current DBS technologies face important limitations, such as large electrode size, invasiveness, and lack of adaptive therapy based on biomarker monitoring. In this study, we investigate the potential benefits of using nanoporous reduced graphene oxide (rGO) technology in DBS, by implanting a flexible high-density array of rGO microelectrodes (25 μm diameter) in the subthalamic nucleus (STN) of healthy and hemi-parkinsonian rats. We demonstrate that these microelectrodes record action potentials with a high signal-to-noise ratio, allowing the precise localization of the STN and the tracking of multiunit-based Parkinsonian biomarkers. The bidirectional capability to deliver high-density focal stimulation and to record high-fidelity signals unlocks the visualization of local neuromodulation of the multiunit biomarker. These findings demonstrate the potential of bidirectional high-resolution neural interfaces to investigate closed-loop DBS in preclinical models.

Deep brain stimulation (DBS) is a widely used neuromodulation therapeutic method to alleviate symptoms and improve the quality of life in individuals with movement disorders such as Parkinson's disease (PD)[1], essential tremor[2] or dystonia[3], and more experimentally, for some neuropsychiatric conditions[4]. Parkinson's disease is characterized by the excessive loss of dopaminergic neurons in the basal ganglia (BG)[5], a group of nuclei critical for movement control and execution. The subthalamic nucleus (STN), part of the BG, is the most common target for DBS in Parkinson's[6]. DBS treatment involves the surgical implantation of a device with electrodes into the STN to deliver high-

frequency electrical stimulation. By modulating neuron firing patterns, DBS aims to normalize excessive inhibitory output from the STN to the thalamus and motor cortex[7,8], thereby reducing motor symptoms such as tremor, bradykinesia and rigidity[9].

However, the use of deep brain implants presents several challenges, including the precise targeting of small deep nuclei in the brain[10], or the mitigation of the tissue damage and inflammation caused by the implantation. Miniaturization of the leads and electrode sites offers an attractive route for reducing the invasiveness and improving the resolution of DBS therapy. However, maintaining low

[1]Catalan Institute of Nanoscience and Nanotechnology (ICN2), CSIC and BIST, Campus UAB, Bellaterra, Spain. [2]University of Manchester, Center for Nanotechnology in Medicine & Division of Neuroscience, London, UK. [3]Instituto de Microelectrónica de Barcelona, IMB-CNM (CSIC), Esfera UAB, Bellaterra, Barcelona, Spain. [4]Biomedical Research Networking Center in Bioengineering, Biomaterials and Nanomedicine (CIBER-BBN), Barcelona, Spain. [5]Bernstein Center for Computational Neuroscience Munich, Faculty of Medicine, Ludwig-Maximilians Universität München, Planegg-Martinsried, Germany. [6]University College London, Queen Square Institute of Neurology, Department of Clinical and Experimental Epilepsy, London, UK. [7]Institució Catalana de Recerca i Estudis Avançats (ICREA), Barcelona, Spain. [8]Institute of Neuroscience, Universitat Autònoma de Barcelona, Barcelona, Spain. ✉e-mail: rob.wykes@manchester.ac.uk; kostas.kostarelos@icn2.cat; joseantonio.garrido@icn2.cat

impedance for high-fidelity recordings and the capability to inject high charge density to achieve tissue activation, remains challenging with the clinically approved electrode materials[11].

Current clinical DBS technology, mainly based on Pt or PtIr metals, uses leads with electrodes on the millimeter scale (typical areas of individual electrodes are 6 mm² or 1.5 mm²). This does not allow to capture of action potential with high SNR, limiting the mapping of a small brain region for precise targeting of the lead and obscuring the observation of multi-unit-based Parkinsonian biomarkers.

Furthermore, the number of electrodes, because of their large dimensions, is low, offering a poor resolution of the stimulation, causing in many cases side effects due to unwanted stimulation of neighboring brain regions or structures[12]. This limitation also affects the adaptability of the electrode array to the expected differences in structure and connectivity of the brain of the different individuals[9], making the treatment less effective.

Another limitation of current DBS is its continuous, open-loop stimulation operation, which can lead to adverse effects, reduced efficacy over time and excessive battery consumption[13]. Closed-loop operation based on the monitoring of electrophysiological biomarkers is a promising strategy to improve DBS[14–16]. Pilot studies have shown limited improvement in clinical outcomes so far, attributed to the lack of reliable biomarkers as feedback to close the loop and suitable electrode technology[14,17]. Closed-loop DBS might benefit from bidirectional electrodes capable of sensing and selectively modulating pathophysiological brain activity at the resolution of single cells, for which new electrode materials and technologies should be developed.

As we demonstrated in a previous study[18], rGO is a high-performing electrode material that exhibits high charge injection and low impedance, which translates into efficient stimulation and recording of brain activity with high fidelity. Aiming to explore the potential of this electrode technology in DBS, in this work we have developed a thin-film device (thickness of 10 μm and a width of 120 μm), equipped with a linear array of reduced graphene oxide (rGO) microelectrodes of 25 μm in diameter. The performance of the device for DBS applications was tested in hemiparkinsonian (induced with stereotactic 6-OHDA toxin administration)[19,20] and healthy control rats. We validated the capability for electrophysiology-guided precise targeting of the STN during insertion, as well as the bidirectional capability for sensing a multi-unit-based Parkinsonian biomarker and for inducing local neuromodulation of the biomarker that lasts temporarily for tens of seconds after stimulation.

Overall these findings underscore the potential of nanoporous graphene-based microelectrodes for high-resolution bidirectional neural interfaces[21]; such technologies could enable real-time monitoring and modulation of brain activity, eventually helping to develop effective neuroelectronic therapies that could be adaptive to the personalized neurophysiological needs of the individual patient.

## rGO thin-film technology and microelectrode performance

In this study, we designed and fabricated flexible microelectrode arrays for recording and stimulation of deep brain structures. The lead is a thin-film device (with a thickness of 10 μm and a width of 120 μm), equipped with 8 microelectrodes (25 μm diameter) spaced 100 μm (Fig. 1a, b and Supplementary Fig. S1). The active material of the electrodes consists of reduced graphene oxide (rGO), connected by gold tracks, and encapsulated on a flexible substrate of polyimide. The stacking of rGO flakes together with oxygen-related vacancies and basal plane defects creates a nanoporous structure which thus enables the preparation of microelectrodes with low-impedance and high-charge injection[18].

The fabricated devices were electrochemically characterized in phosphate buffer saline (PBS) electrolytes to evaluate their perfect functionality. The charge injection limit was determined by the

maximum injected current that leads to voltage polarization within the rGO water window (−0.9 V to 0.8 V, Supplementary Fig. S3a). Considering typical stimulation protocols used in DBS applications, biphasic current pulses of 100 μs were injected while recording the voltage at the electrodes (Fig. 1c). For the 25 μm diameter rGO microelectrodes, currents as high as 100 μA can be safely injected (Fig. 1d), resulting in a charge injection limit (CIL) of 2.3 mC/cm². This value is significantly higher than for conventional metal microelectrodes[22], and within the range of CIL of novel electrode materials used for similar applications[23,24]. Electrochemical impedance was also measured in PBS to determine the charge transfer properties of the electrodes. Figure 1e presents the full spectrum of the impedance, showing the average values of magnitude and phase for the 8 electrodes of a single array. Figure 1f summarizes the average magnitude at 1 kHz and 1 Hz for the electrodes of ten different arrays ($29.4 \pm 5$ kΩ at 1 kHz and $5.2 \pm 0.6$ MΩ at 1 Hz, total average $n = 79$). Compared to flat metallic electrodes, rGO microelectrodes exhibit low impedance due to the increased electrochemically active surface area that results from their nanoporous structure[25]; similar values are reported for microelectrodes based on PEDOT or IrOx, which are also characterized by a high electrochemically active surface area[26,27]. The stability of the microelectrodes was assessed during continuous electrical stimulation with biphasic current pulses (100 μs pulse width and 50 μA current amplitude, corresponding to 1 mC/cm²); after 100 million pulses, the microelectrodes did not exhibit any significant change in their characteristics (see Supplementary Fig. S2), in good agreement with previous work[18].

## Implantation of flexible arrays

The use of neural interfaces based on μm-thin flexible substrates has been reported to be able to mitigate foreign body response of the tissue[28,29]. However, the implantation of such thin leads poses a significant challenge, particularly when inserting the lead deep into the brain. To address this issue, we used a strategy in which the flexible lead is temporarily attached to a rigid shuttle by a dissolvable adhesive polymer (Supplementary Fig. S3c, d). The rigid shuttle provides sufficient stiffness to penetrate the neural tissue during the implantation procedure and can be separated from the flexible array once the biocompatible adhesive polymer is dissolved (Fig. 1g and Supplementary Fig. S4a). We used a silicon shuttle of 80 μm in width and 60 μm in thickness. These dimensions were chosen to offer sufficient rigidity for a cm-deep insertion without fracturing, while minimizing tissue damage[30]. To validate the penetration capability of the shuttle, we measured the buckling force generated by contacting the shuttle on a rigid surface (see Supplementary Fig. S5c for details). Experiments revealed a buckling force of $37.9 \pm 1$ mN, which is above the reported force threshold required to penetrate the rat dura with implants of similar dimensions[30].

The speed of insertion is another critical factor for mitigating brain injury, preserving nearby neurons, and ensuring the capacity to record single-neuron activity[31]. We followed a two-step method that involves an initial rapid insertion (100 μm/s) followed by a slow insertion (5 μm/s) around the structures of interest, starting at 6 mm deep. Measurements of the insertion force in agar (see Supplementary Fig. S5b) revealed that the first step at high speed is characterized by force peaks of 20 mN[32]. However, during the second step, conducted at 5 μm/s, no stretching or compression forces could be measured (force resolution of the experimental is ∼1 mN), considered relevant to reducing cell damage in the brain[31].

To allow the removal of the shuttle following the insertion of the probe to the desired depth, we investigated dissolvable adhesive polymers that could remain stable during insertion and then slowly dissolve by interacting with brain fluids. Given the constraints of our implantation procedure, the adhesive polymer shall be stable for at least 10 min (the insertion time), and its stability should not be

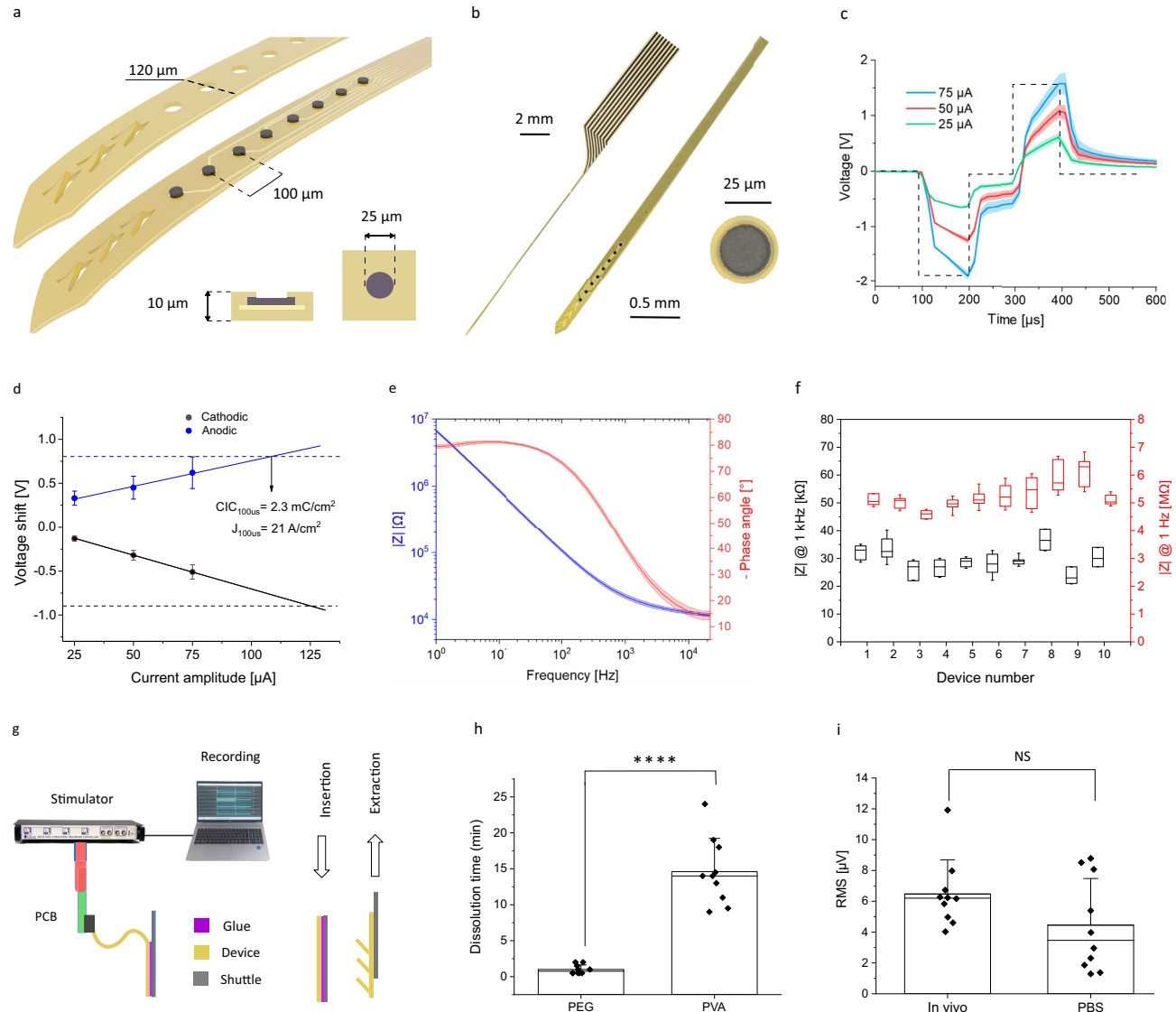

**Fig. 1 | Bidirectional graphene-based DBS technology. a** Illustration of microelectrode linear array at the tip of the device, with top and bottom polymeric layers. The device cross-section displays the layers of polyimide, gold, and reduced graphene oxide. **b** Photograph of the device, with a zoom of the tip and of a microelectrode. **c** Average voltage polarization (solid lines), with shadowed area for standard deviation (STD) ($n = 8$), of the DBS protocol, consisting of biphasic current pulses (100 μs pulse width) and increasing current levels. **d** Average voltage drop at the electrode-electrolyte interface with STD ($n = 8$). The dashed line represents the safe voltage limits for the anodic (blue) and cathodic (black) polarizations, from which the charge injection limit can be calculated (CIL = 2.3 mC/cm²). **e** Average impedance magnitude (blue) and phase (red) ($n = 8$). **f** Impedance magnitude at 1 Hz (red) and 1 kHz (black) of the microelectrodes for 10 different arrays; boxplots represent 25 and 75 percentile, with the median value line and whiskers for the STD (for each device $n = 8$). **g** Schematic of the in vivo set-up for bidirectional recording and stimulation. The flexible device is attached to a rigid shuttle with a dissolvable adhesive polymer. Once the insertion is completed, the flaps at the device's end keep the device in place during the extraction of the shuttle. **h** Average dissolution time in agarose at 38 °C of the polyethylene glycol (PEG, $n = 10$) and polyvinyl alcohol (PVA, $n = 10$). **i** RMS of the voltage amplitude of the signal recorded (25 s, in the range 200–2000 Hz) in PBS before insertion and in the brain after reaching the STN. The plot displays the average values for the electrodes of 10 different devices (for each device $n = 8$).

overextended more than necessary (<30 min), to minimize the surgery duration and facilitate post-implantation recovery of the animal. Polyethylene glycol (PEG) is a biocompatible adhesive polymer that has been used in previous studies[33–35] to temporarily fix flexible probes to stiff shuttles for the implantation of intracortical devices. We tested PEG to glue our devices to the silicon shuttle, mimicking the experimental in vivo conditions (agar phantom with an ion concentration of 290 mM, at 38 °C). We found an average dissolution time of merely $1 \pm 0.6$ min, which is considerably lower than the previously reported[36], and which we attribute to the challenging conditions of our experiment, in particular the very small width of the flexible array. For such narrow devices, the dissolution time of PEG, thus, is not enough for the

deep and slow insertion required to reach deep brain structures such as the STN[37].

To meet these specifications, we prepared a biocompatible, water-soluble polyvinyl alcohol (PVA) adhesive polymer (see Supplementary Information for details on the PVA preparation). The average dissolution time of the PVA-based adhesive polymers results in $14.6 \pm 4.6$ min (Fig. 1h), which is compatible with the duration of our insertion protocol and, at the same time, with a short post-insertion dissolution time.

Shuttle removal introduces an additional challenge since friction at the shuttle-polyimide interface could eventually displace the implanted thin and flexible device. Given that the rat's STN is a very

small structure (150–200 μm anteroposterior, 100–150 μm medio-lateral and 200–250 μm dorsoventral), a relatively small displacement of the device could eventually move the microelectrodes outside the STN region. To prevent this, we implemented an anchoring system into the internal structure of the devices (Fig. 1a), as previously used in similar nerve implants[36]. This system features internal flaps that flex during extraction but do not affect the insertion procedure (Fig. 1g).

To validate that the insertion procedure does not affect the functionality of the microelectrodes, we compared the root mean square (RMS) of the signal recorded (200–2000 Hz) with electrodes before implantation (measured in PBS) and after completing the insertion of the devices in the brain (see Fig. 1i). The results confirm that the microelectrodes maintain their functional integrity after implantation, and thus are not damaged by the implantation procedure.

## Recording capability and high-precision mapping of STN

The low impedance of the rGO microelectrodes is expected to translate into high-fidelity detection of neural signal[38]. To evaluate the recording capability in deep brain structures, we conducted acute in vivo experiments in which the linear arrays were implanted in anesthetized rats, and neural activity was recorded with electrodes inside and outside the STN. A representative signal recorded with an electrode inside the STN, depicting traces that correspond to the signal filtered in different frequency bands, shows the capability of the rGO microelectrodes to monitor high-fidelity neural activity from slow local field potentials (LFPs) up to fast multi-unit activity (MUA) (Fig. 2a); for comparison, we show the postmortem recording of the same electrode. Figure 2b depicts the average power spectral density (PSD, $n = 8$ electrodes) of the in vivo and post-mortem recording time shown in Fig. 2a. The ratio between both PSDs, which can be used as a figure of merit (performance metric) for the signal-to-noise ratio (SNR)[39], is above 10 for all the frequency bands. In the 200–2000 Hz frequency range, where MUA is typically assessed, it was obtained (Fig. 2d) an average spike-to-noise ratio ($SNR_{spike}$) of $10.4 \pm 3.2$ ($n = 9$ rats), in line with what can be achieved by high-performance electrode materials[40,41].

In a proof-of-concept chronic study (see Supplementary Fig. S6), we found that both LFP and spike activity were recorded in an awake rat with SNR > 10 after 3 weeks of implantation. The capability of recording spiking activity with low noise can be correlated to the evolution of the electrode impedance over time. We conducted in vivo impedance measurements of the microelectrodes in chronically implanted arrays. Supplementary Fig. S7a shows the average value of the module the impedance measured at 1 kHz for the working electrodes of four different arrays implanted in the STN; the average impedance magnitude measured in vivo ($n = 31$), around $169 \pm 52$ kΩ at 1 kHz, is higher than the impedance measured in PBS (Fig. 1f), due to the lower conductivity of the neural tissue[42]. In the proof-of-concept chronic study, the in vivo impedance remained stable over the course of three weeks (Supplementary Fig. S7b), suggesting no significant degradation of the electrodes or changes at the electrode/tissue interface[43].

The high-fidelity recording capability of the rGO microelectrodes, combined with the high density of microelectrodes in the linear array, allows the precise localization of the implanted device with respect to the STN during the implantation. This is enabled by the characteristic intense spiking activity of the STN compared to the activity of adjacent brain regions like the zona incerta (above the STN) and the internal capsule (below the STN), which are both relatively silent[44,45]. A representative recording of brain activity obtained from an implanted array targeting the STN is presented in Fig. 2d.

The activity recorded with the electrodes is presented in two sets of curves, corresponding to the signal filtered in the LFP band (1–200 Hz) and in the MUA frequency band (200–2000 Hz). While the LFP signals appear uniform across all electrodes, both in time series and with spectral analysis (see Supplementary Fig. S8), the MUA signals reveal clear differences in the spiking activity recorded by the different electrodes. The two more distal electrodes in the array exhibit a significantly higher spike rate than the rest of the electrodes (see Fig. 2d–f and discussion below), indicating that these two electrodes are located within the STN. To assess the suitability of the two frequency bands (LFP and MUA) for localization of the STN, we computed the cross-correlation (see Methods) between the signal recorded with the more distal electrode in the array and each one of the others, for the two frequency bands. The analysis confirms a very high correlation (>0.9) for all the electrodes in the LFP band; for the MUA band, on the other hand, the high correlation is only observed for the two more distal electrodes, whereas for the rest of the electrodes, the cross-correlation is below 0.4 (Fig. 2e).

The raster plot (Fig. 2f, left) illustrates the spike count for each electrode of an array, from which we can then define and quantify a color-coded spike rate histogram (Fig. 2f, right); see Methods for calculation of color-coded spike rate. These results suggest that we can place 2 or 3 electrodes (the ones with a high spike rate) within the tiny STN structure (200–250 μm along the dorsoventral direction), while the rest of the electrodes are located outside the STN. We conducted this analysis across multiple implantations by plotting the color-coded spike rate for each electrode in arrays implanted in eight different animals (Fig. 2g). The shadowed region (violet) marks the position of the two electrodes with the highest spike rate of each array, which we use to tentatively identify the position of the STN in each of the experiments.

## Monitoring of biomarkers in the STN of hemiparkinsonian rats

To explore the suitability of the rGO microelectrode technology for monitoring Parkinson's disease biomarkers, we used the well-established 6-hydroxydopamine (6-OHDA) rat model, in which dopaminergic neuron depletion is selectively induced by unilateral administration of 6-OHDA into the medial forebrain bundle[20]. The generated asymmetric neurodegeneration can be evidenced by a marked reduction in dopamine fluorescence within the targeted hemisphere. Figure 3d visually demonstrates a dopamine deficit in the hemisphere injected with the neurotoxin, compared to the other brain, injected with saline, which is dopamine-intact. This dopaminergic cell loss was statistically quantified by computing the ratio of tyrosine hydrolase pixel intensity in the striatum of the injected side over the non-injected one (Fig. 3a), and it results lower for the PD group ($0.44 \pm 0.11$, $n = 6$) compared to the control one ($1.09 \pm 0.1$, $n = 4$).

Before implantation of the devices, the animals underwent a series of behavioral tests (see details in Methods section) to assess their response to the neurotoxin injection. In contrast to the control rats treated with saline, the rats injected with 6-hydroxydopamine exhibited a marked asymmetry in paw usage[46]. Notably, PD rats were unable to effectively use the left paw when the right hemisphere was injected with the neurotoxin. This behavioral change was quantitatively evaluated by placing each rat in a transparent cylinder and recording the number of times it touched the cylinder wall with each paw (forelimb asymmetry test, Fig. 3b). The analysis of the asymmetry ratio (see Methods), revealed significant differences for the PD and control groups, $0.72 \pm 0.24$ and $0.14 \pm 0.1$, respectively. The second behavioral assessment involved administering apomorphine to the rats and observing their rotational behavior in the cylinder (apomorphine-circling test, Fig. 3c). Typically, rats affected by the 6-OHDA injection display an excessive number of rotations, attributed to the asymmetry in their response to the movement induced by apomorphine. The average number of rotations per minute was significantly higher in the PD group $6.35 \pm 4$, compared to the control

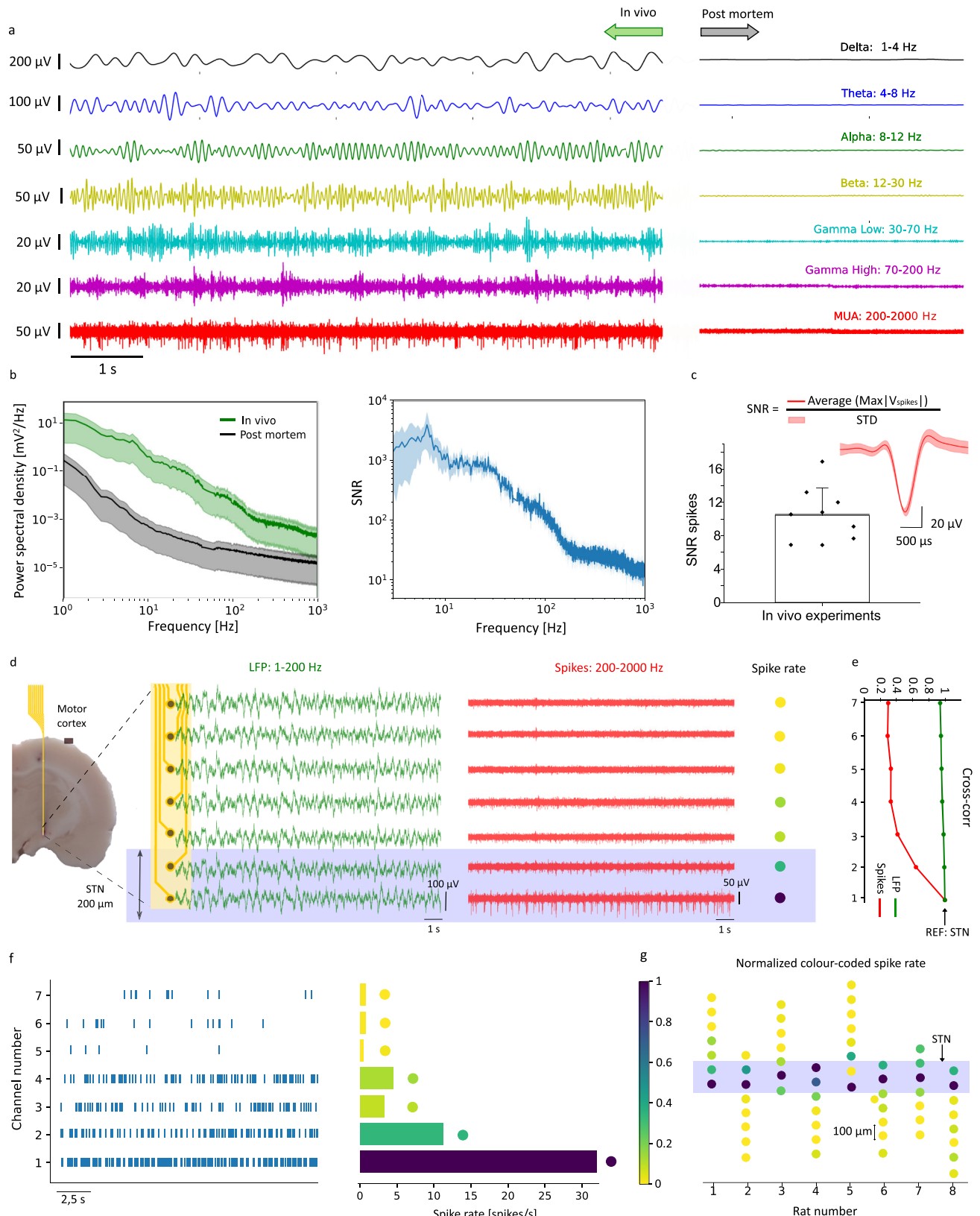

one $0.1 \pm 0$ (Fig. 3c). Both behavioral tests confirm that the animals treated with 6-OHDA exhibited sensory-motor anomalies characteristic of the Parkinsonian rat model, in clear contrast with the observations for the healthy control ones. Rats that completed more than 4 rotations per minute were considered hemiparkinsonian[47] and thus suitable for implantation.

Animals of PD and control groups were unilaterally implanted (see details in the Methods section) with the devices, with the objective of recording the neural signal from the electrodes in the STN and evaluate the capability of identifying electrophysiological biomarkers of the hemiparkinsonian rat model. Figure 3d, e shows characteristic recordings (MUA band) for electrodes in the STN of a PD rat and of a

**Fig. 2 | Recording capability of rGO microelectrodes and STN localization by high-density microelectrode array. a** Representative brain activity recording of one electrode located in the STN, filtered for different frequency bands, for in vivo (left) and post-mortem (right) conditions. **b** *Left*: Average power spectral density (PSD), with STD, of the signal recorded with the electrodes of one array ($n = 8$ electrodes), comparing in vivo (green) and post-mortem (black) conditions. *Right*: Average signal-to-noise ratio (SNR), with STD, calculated from the ratio between in vivo and post-mortem PSDs in the left panel. **c** Average spike-to-noise ratio (SNR$_{spike}$)over 100 s, for the electrodes located in the STN. Each dot represents one acute experiment (PD = 6 and control = 3 rats). The inset shows an example of the average spike shape of a PD rat (the shadow area represents the STD of the spikes recorded over 100 s). **d** Representation of brain activity recorded with the 8 electrodes of one array, also depicting an image of the brain histology superimposed with a schematic of a device in the STN. The two groups of traces correspond to the recordings of each of the electrodes of the array, filtered in the LFP (green) and MUA (red) range. The blue rectangle highlights the traces corresponding to electrodes in the STN. The color-coded spike rate quantifies the number of spikes for the different electrodes of the array. **e** Cross-correlation between channel 1, located in the STN, and each one of the other electrodes, quantified at lag 0 for both LFP (green) and spikes (red). **f** *Left:* Raster plot of the detected spike events over time for each electrode of the array shown in panel (**d**); *Right:* Color-coded spike rate histogram of the raster plot, in which spike events are quantified over 100 s. **g** The color-coded dots represent the average spike rate (in the recording period of 100 s) of the electrodes of the arrays implanted in 8 different animals. The color codes are normalized by the higher spike rate channel of each array. The blue rectangle represents the expected position of the STN.

control one, respectively. The recordings from the PD and control rats reveal clear neural spikes, but there are significant differences in the spike activity for both animal groups in terms of spike rate and firing pattern. We calculated the total spike rate of the activity recorded by electrodes in the STN and found that the average spike rate for the control group was $15.3 \pm 7.6$ spikes/s, compared to $32.3 \pm 12.7$ for the PD group (Fig. 3f), consistent with the expected enhanced neural firing activity of the hemiparkinsonian rats[48]. Besides differences in the spike rate, the 2 animals present different firing patterns (Fig. 3d, e): while the recording in the control rat exhibits a characteristic tonic spiking activity, the recording of the PD rat is characterized by a prominent bursty activity as previously reported[48–50]. The analysis of all in vivo experiments was consistent with the exemplary recordings in Fig. 3d, e, as illustrated in Fig. 3g, which summarizes the calculated number of events per second for both groups (an event is defined as a single tonic spike or a burst of spikes). While the PD group exhibited an average rate of approximately $7.1 \pm 3.5$ bursts/s, in the control one it was lower than 0.1 bursts/s. These bursts are groups of $3 \pm 0.6$ spikes on average (Supplementary Fig. S8b) which fire very close to each other, usually around 400 Hz (see Fig. 3i). In the case of the tonic spike pattern, similar rates were obtained for both groups, with values around 15 spikes/s (see Fig. 3g). This indicates that while the tonic spiking pattern is consistent across both groups, the PD group exhibits an additional bursty component, which contributes to the overall difference in total spike counts.

To further assess differences in the firing activity between the 2 groups, we calculated the interspike frequency (IF), which is defined as the inverse of the interspike time interval. Figure 3h plots the histogram of the interspike frequency for the experiments shown in Fig. 3d, e, considering a normalized number of counts. The IF distribution for the PD rat is bimodal, with the low-frequency peak corresponding to tonic activity and the high-frequency peak associated with the burst-type activity (Fig. 3h). In contrast, the IF distribution of the control animal is unimodal, characterized by a single tonic-related low-frequency peak. The frequency peaks of the IF distributions of the 2 groups are summarized in Fig. 3i.

In contrast to the difference in spike firing patterns observed in the PD and control animals, the analysis of the recordings in the LFP bands did not reveal any significant difference between the two groups. The average power spectral density (PSD) of the signals recorded with electrodes located in the STN, shows that the PSDs do not allow the identification of any statistically significant difference between the PD and control groups (Fig. 3j). A similar outcome can be obtained from the analysis of the median power of the recorded signal for each frequency band for both groups (Fig. 3k). This analysis suggests that, contrary to the case of LFP signals, MUA is more effective in encoding PD disease-related biomarker information in anesthetized hemiparkinsonian rats.

## DBS neuromodulation with rGO microelectrodes
After validating the suitability of the high-density linear array of rGO microelectrodes for localizing deep brain structures like the STN, as

well as for identifying PD-related high-frequency electrophysiological biomarkers, we next investigated the effect produced by focal electrical stimulation on PD and control animals. To this end, we applied electrical stimulation consistent with "standard" high-frequency DBS protocols and monitored its effect by comparing neural activity before and after stimulation. In particular, we used biphasic current pulses of 100 µs in pulse width, with various amplitudes and frequencies. A 1-minute stimulation protocol (current amplitude of 75 µA and frequency of 130 Hz) changes the spike firing pattern in a PD animal (Fig. 4a). Prior to the stimulation protocol, the recorded signal exhibits the spike firing characteristic of the PD group, visible in the plot of the interspike frequency evolution over time (Fig. 4a) and in the corresponding interspike frequency histogram (Fig. 4b), revealing the previously discussed bimodal distribution, with tonic and burst spiking pattern. After stimulation (the stimulation was performed with the same microelectrode used for recording), the recorded activity shows a clear reduction of the burst activity: the interspike frequency distribution over time shows that the burst activity, represented by the high-frequency band, is significantly reduced and almost suppressed for several tens of seconds after the stimulation protocol; at this point, the brain activity looks normalized, similar to the activity of a healthy control rat (Supplementary Fig. S10a, b). The effect of stimulation on brain activity is better visualized by comparing the histograms of the normalized IF distributions pre- and post-DBS, as shown in Fig. 4b. The normalization facilitates the evaluation of the change in frequency distribution, more than the variation of the total spike rate (not normalized IF distribution pre- and post-DBS for both PD and control groups are shown in Supplementary Fig. S9). After stimulation, the amplitude of the burst peak is reduced to half its value before stimulation, while the IF distribution peak related to the tonic activity is shifted to lower frequencies (Fig. 4b). When applying the same DBS protocol to a rat of the control group, no significant change in the spike firing pattern is observed (Supplementary Figs. S9b and S10); the IF distribution associated with the tonic activity is unaffected both in its peak frequency position and width. mV²/Hz.

We extended the experiment to the rest of the animals of the PD group; the summary of the analysis is depicted in Fig. 4c. In particular, we calculated the ratio (R), post- to pre-DBS of different spiking-activity related metrics: the total spike rate, the tonic spike rate, the burst rate, and the burst spike count (Fig. 4c). The total spike rate and the tonic spike rate show certain variability in the response to the electrical stimulation. This variability could be explained by the coexistence of excitatory and inhibitory neurons within the STN, which can be either excited or inhibited by the stimulation[51]. Interestingly, while the tonic spike rate and total spike rate may increase in some cases after stimulation, the burst rate consistently decreases after stimulation for all investigated PD animals, evidenced by the median value of the burst rate ratio (R = 0.29). This finding suggests that the applied stimulation does not merely suppress neural activity but contributes to the desynchronization of the spike firing activity.

Various stimulation protocols were explored to assess the capability of modulating burst activity at different frequencies and current amplitudes, as detailed in the Supplementary Information (Supplementary Fig. S11). The analysis shows that, for the case of the 130 Hz stimulation protocol, a current amplitude of 25 µA is not sufficient to change the burst-type activity (10% burst reduction); in

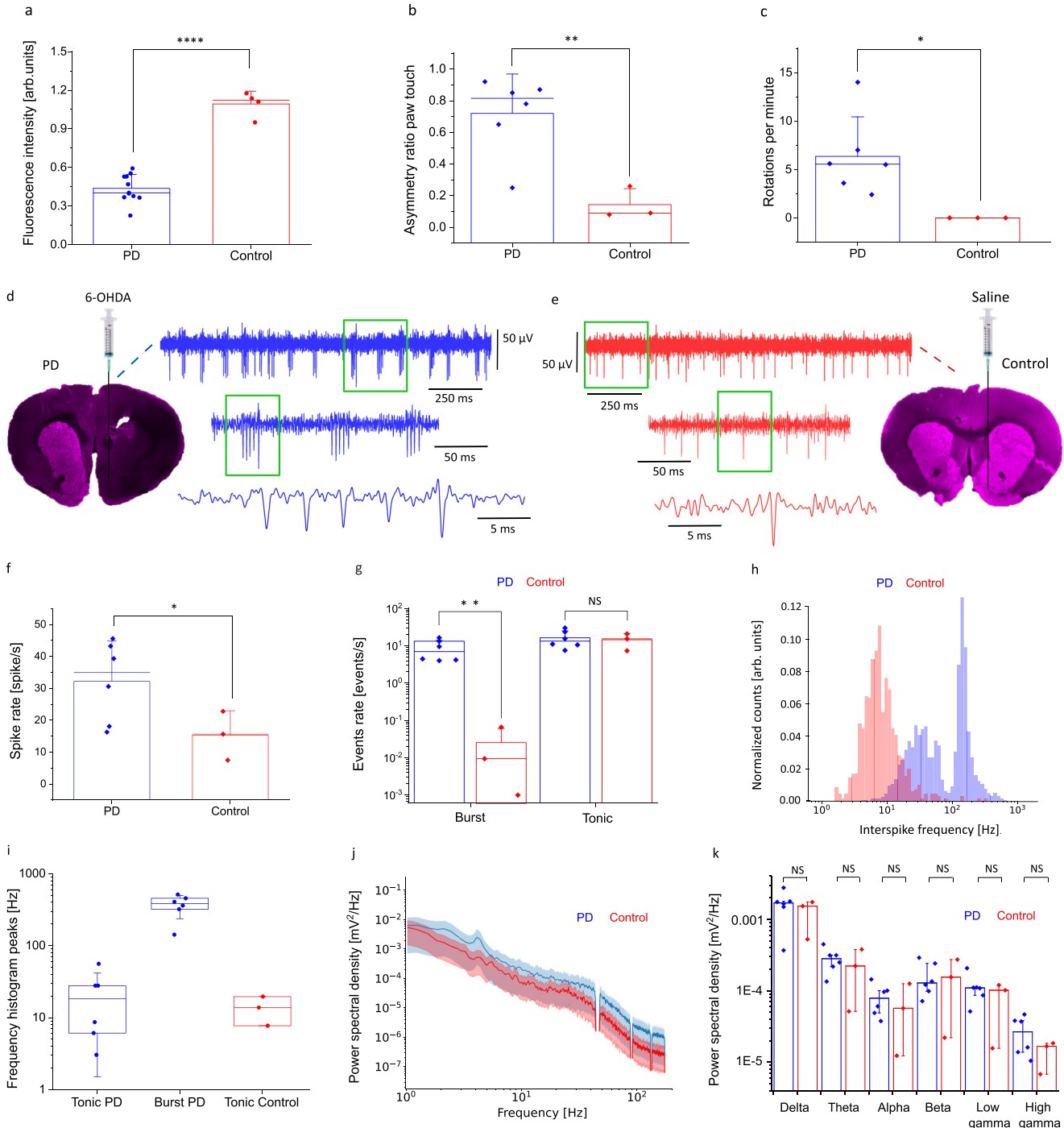

**Fig. 3 | Electrophysiological biomarkers recorded in the STN. a** Tyrosine hydrolase (TH) expression in the striatum as a measure of the extent of dopaminergic cell depletion. The average fluorescence intensity is expressed as a percentage of the contralateral side (animals analyzed: PD = 10, control = 4). **b** Forelimb asymmetry test, quantifying the average asymmetric ratio in the use of the left and right paw (PD = 6, control = 3). **c** Apomorphine-circling test, quantifying the average number of rotations per minute after epimorphine injection (PD = 6, control = 3). **d, e** Exemplary recordings (MUA band, 200 Hz–2 kHz) of electrodes in the STN of the PD and control rats together with fluorescence images of brain tissue slices. **f** Average spike rate (over 100 s) with of one electrode in the STN of the two groups (PD = 6, control = 3). **g** Average number of spiking events, defined as bursts per second and tonic spikes per second, detected over 100 s (PD = 6, control = 3). **h** Interspike frequency distribution, normalized by the total number of counts in 100 s, calculated from recordings of a PD (blue) and a control rat (red). **i** Frequency peaks of the interspike frequency histograms for the 2 animal groups (PD = 6 in blue, control = 3 in red). The box plot represents the 25 and 75 interquartile. **j** Comparison of the average PSD with STD (shadow area) of the signal recorded in the LFP range (1–200 Hz) over 55 s for the two animal groups (PD = 6, control = 3). **k** Quantification of the total power over 100 s for each frequency band (delta 1–4 Hz, theta 4–8 Hz, alpha 8–12 Hz, beta 12–30 Hz, low gamma 30–70 Hz and high gamma 70–200 Hz), showing the median value with 25 and 75 interquartile whiskers (PD = 6, control = 3).

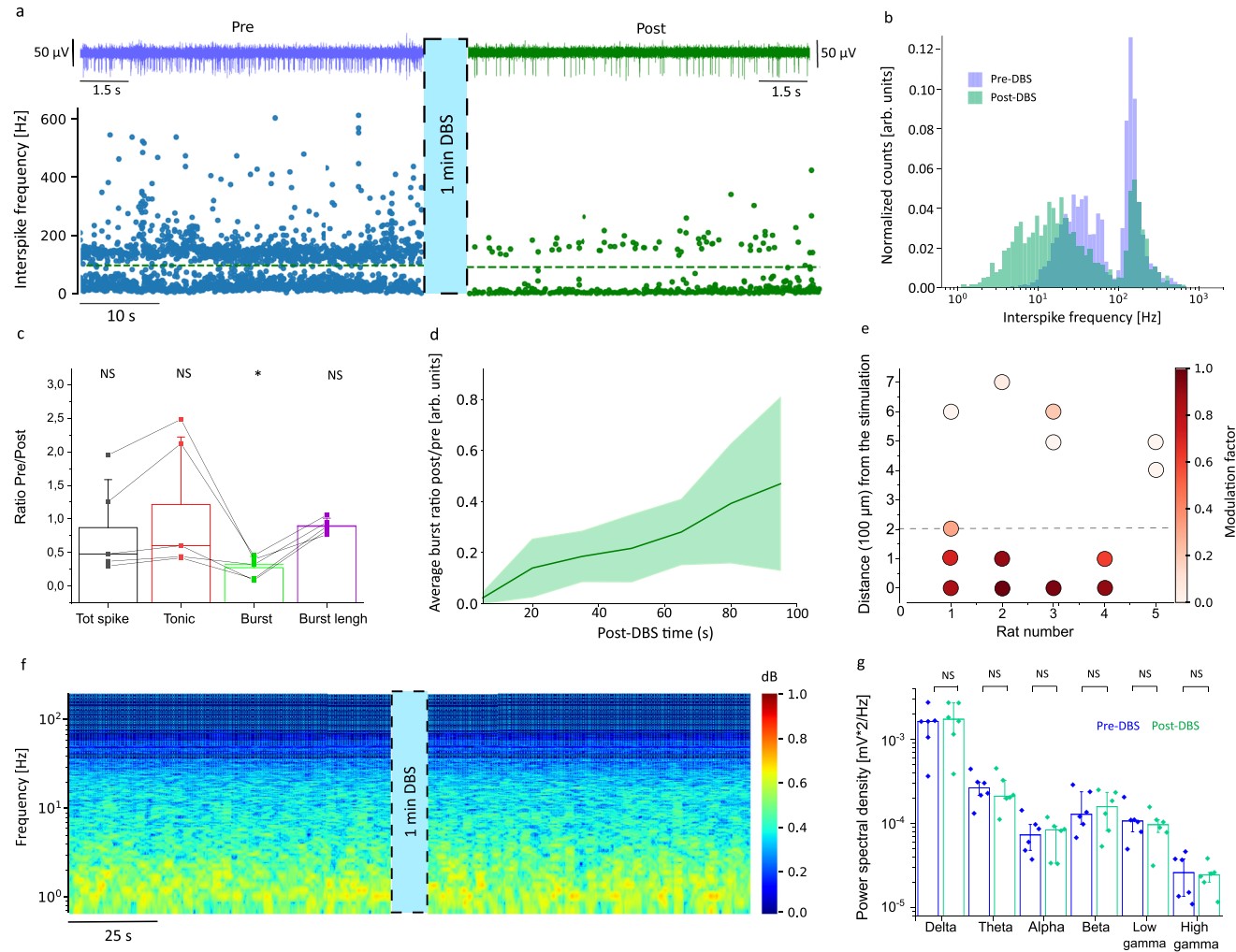

**Fig. 4 | Deep brain neuromodulation in the STN of Parkinsonian rats. a** Effect of microelectrode stimulation on the spike firing activity; the same microelectrode was used for stimulation and recording. The traces above correspond to the recordings (200 Hz–2 kHz) with a microelectrode in the STN of a PD animal, pre (blue trace) and post (green trace) stimulation. Below are shown corresponding time-dependent interspike frequency plots, pre and post-DBS; the green dashed line (at 100 Hz) indicates the division between the tonic and burst activity. **b** Histogram of normalized IF spike distribution calculated pre- and post-DBS in a PD animal; each histogram quantified over 100 s. **c** Average ratio (post- to pre-DBS) of different spiking-activity metrics: total spike rate, tonic spike rate, burst rate, and burst spike count (number of spikes per burst), calculated for the PD group

(PD = 5). **d** Time evolution of the post-pre ratio of burst rates, depicting average (solid line) and STD (shadow color) for the PD group, quantified each 15 s. **e** Color-coded modulation factor of the burst activity (definition in main text), displaying the impact of the distance of the stimulating microelectrode to the STN. The dashed line indicates the threshold distance at which the modulation factor drops to 0.5. Each dot corresponds to the modulation factor measured (over 100 s?) in one electrode, for 5 different arrays (PD = 5). **f** Normalized spectrograms of LFP (1–200 Hz) activity of a PD rat, pre and post-stimulation. **g** Total PSD of recorded signal, quantified (55 s) in different LFP frequency bands. Data were calculated for the PD group (n = 6), showing the median value with the corresponding 25 and 75 interquartile error bars.

contrast, current amplitudes of 50 µA and 75 µA are effective in modulating neural activity (95% burst reduction, Supplementary Fig. S11c, d). The minimum current for which modulation of neural activity is observed corresponds to a charge density of 1 mC/cm² (considering the pulse width of 100 µs and the electrode diameter of 25 µm); this current level corresponds to a charge injection limit that cannot be reached by flat metal microelectrodes and is only compatible with a few electrode materials[24,26,52,53]. We also studied the effect of the stimulation frequency (see Supplementary Fig. S11a, b). The current was maintained at 75 µA while varying the stimulation frequency. No modulation was noted at 10 Hz, partial reduction of burst activity was observed at 40 Hz, and very effective modulation was achieved at 130 Hz (>95% reduction).

The long-term modulation capability of the rGO microelectrodes was demonstrated in a proof-of-concept experiment. After 3 weeks,

the electrodes could effectively modulate high-frequency brain activity (see Supplementary Fig. S5e). Importantly, with a charge density of 1 mC/cm², the voltage shift at the electrode-tissue interface was still within the electrochemical safe limit of rGO. This result complements the data obtained in previous work[18], showing the recording of auditory evoked potential in mice for 3 months and the capability to inject electrical stimulation in the sciatic nerve and induce muscle activation for 2 months. While these results indicate a stable electrode behavior, further investigation is needed to understand the effect of high current density on brain tissue.

Overall, it is important to consider that the observed modulation of neural activity, following stimulation, is not permanent and the burst activity is gradually recovered in the time scale of hundreds of seconds, indicating a reversible process. Figure 4d quantifies the recovery process, presenting the average burst rate ratio, post- to pre-

DBS, over time for the PD group. This analysis reveals that immediately following DBS treatment (20 s after) burst activity is almost completely suppressed (less than 5% remains), slowly recovering 40% of its original level after 90 s. A more detailed description of the evolution of the burst recovery after stimulation is presented in Supplementary Fig. S12. The evolution of spiking activity post-DBS involves a gradual decrease in the tonic spike rate and a concurrent increase in the burst rate (Supplementary Fig. S12). Similar studies noted a general recovery process of the total spike rate after DBS[54].

We then assessed the capability of the rGO microelectrodes for focal stimulation[55] by investigating the efficacy of modulating brain activity at increasing distances. To this end, we quantified a "burst modulation factor", $F_{burst}^{mod}$, which represents the change in the burst activity induced by the stimulation, defined as $F_{burst}^{mod} = \frac{Burst_{pre} - Burst_{post}}{Burst_{pre}}$, where $Burst_{pre}$ and $Burst_{post}$ correspond respectively to the burst rate before and after DBS. It is important to note that, while pre-DBS burst activity is relatively constant over time, post-DBS burst activity increases over time (Fig. 4d), implying that $F_{burst}^{mod}$ is an averaged estimation of a dynamic phenomenon. The $F_{burst}^{mod}$ is quantified, with a color-coded scale, for several electrodes of different arrays ($n = 5$ PD animals), considering an absolute distance from the stimulating electrode to the recording sites (Fig. 4e). This analysis confirms that a very effective modulation is recorded in the electrode used for stimulation ($F_{mod}^{burst} \cong 1$). For microelectrodes located 100 μm "away" from the stimulating one, $F_{burst}^{mod}$ already drops below 0.8. For microelectrodes located 500 μm away, $F_{burst}^{mod}$ drops below 0.2, meaning almost no effect of modulation. These results suggest that focal modulation of the STN can be achieved, provided that the device is precisely positioned within the target area.

We also assessed the impact of stimulation on the LFP signal; the analysis did not evidence any significant change in brain activity for any of the frequency bands from 1 to 200 Hz, as illustrated by the spectrogram in Fig. 4f. To confirm this result, we compare the power of the recorded LFP signal in each frequency band pre- and post-DBS, and it does not reveal any significant change in the LFP activity upon stimulation (Fig. 4g). The lack of sensitivity of the LFP signal to the stimulation is attributed to the focal stimulation provided by the rGO microelectrodes[55], which are capable of modulating local neural activity without altering the less local, LFP-encoded neural activity, in particular under the anesthesia conditions of our experiments, characterized by highly synchronized LFP activity.

## Discussion

In order to expand DBS adoption, it is key to reduce the invasiveness of the therapy, minimize unwanted side effects by increasing stimulation precision, and enhance efficacy enabling adaptive modulation of brain activity. A new generation of electrode materials, offering better performance and bidirectionality, is needed to help resolve some of these challenges. As we reported in a previous study[18], reduced graphene oxide is a high-performing material due to the nanoporous structure that increases the electrochemically active surface area, resulting in low-impedance and high-charge injection capacity. Besides its state-of the art electrochemical and mechanical properties, rGO is also a relatively low-cost material, can be integrated with flexible microelectronics and microfabrication processes, and it is highly biocompatible[21,56] and chemically inert, which makes it promising for neural interfaces.

In this work. we investigated the potential benefits of using rGO technology for deep brain stimulation in Parkinson's disease. The devices were designed with a high-density linear array of 8 microelectrodes of 25 μm in diameter, spaced 100 μm, in a compact and lead 120 μm wide and 10 μm thin. For the implantation of the flexible leads, we have developed a methodology that uses an optimized dissolvable adhesive polymer (PVA) to temporarily attach a rigid and thin silicon shuttle, that enables a slow and controlled insertion while minimizing

brain damage. The leads are equipped with an anchoring system to keep the electrodes in place during the removal of the shuttle, which is crucial to deliver electrical stimulation in the correct place and minimize the side effects of off-target stimulation. The structural and electrical properties of the rGO material allow for the miniaturization of the electrodes to microscale, while maintaining the capability to record single-neuron activity with high SNR and to inject enough charge to induce neuromodulation.

The combination of small electrodes with high-fidelity recording capability facilitated the precise localization of the STN by monitoring the specific pattern of spiking activity of these brain nuclei. Millimeter-size electrodes, as currently used in the clinic, do not allow to easily provide information about the lead position inside or outside the STN, since these large electrodes cannot monitor spiking activity, making targeting or localization based on electrophysiological recordings in the LFP range not possible with the same device.

Our study with hemi-parkinsonian and control rats conducted under anesthesia revealed the presence of a multiunit-based Parkinsonian biomarker. The brain activity in the STN of Parkinsonian rats exhibits a unique spike distribution characterized by a combination of tonic and burst patterns. In contrast, the control rats display only a tonic pattern. This brings the possibility of using an additional biomarker for closed-loop, adaptive DBS in Parkinson's disease, different from the more standard beta peak in the LFP range[57], whose use in the clinic has been recently questioned[16,58,59].

The bidirectional property of recording and stimulation with the same electrode allows us to investigate local effects produced by the electrical stimulation on this biomarker. We explored several high-frequency DBS protocols and monitored the effects on the neural activity of hemiparkinsonian and control animal groups. By analyzing STN spiking distributions before and after stimulation we confirmed a reduction of bursting activity in the STN of the PD animals, while the tonic spiking activity increases in some cases, indicating a modulation process, rather than a global suppression effect. Compared to other studies where no post-DBS modulation of the burst activity is reported[60,61], our study indicates that bidirectional high-resolution neural interfaces facilitate the visualization of local neuromodulation effects in the high-frequency activity that lasts temporarily for tens of seconds after stimulation. This modulation can be achieved only by injecting a high-density charge of 1 mC/cm² (for a"standard" DBS protocol of 100 μs pulse width at 130 Hz) which cannot be provided by flat metal microelectrodes and is possible only with few electrode materials[24,26,52,62]. Moreover, delivery of DBS in the STN of hemi-parkinsonian rats normalizes the spiking activity to a state similar to the control rats, suggesting a potential therapeutic effect that will be evaluated in future studies. Interestingly, no significant change was observed in the LFP range after DBS, presumably due to the high focality of the treatment[63] (approximately 200 μm) which does not modulate LFP-encoded neural activity, especially under anesthesia, since the brain signal is highly synchronized.

In summary, this work demonstrates the benefits of using bidirectional rGO microelectrodes for improved acquisition of local electrophysiological biomarkers such as STN bursting activity, and the capability to modulate them with high efficacy. This opens the potential use of this technology to investigate the mechanisms around DBS and to implement closed-loop therapies where stimulation parameters are either adaptive or responsive to the biomarker level. Future translation of this technology to humans will face important challenges such as the implementation of a new insertion strategy[64,65] compatible with the clinical surgical procedures and a lead design adapted to the human brain. Additionally, it will be necessary to achieve chronic recording of spiking activity over years of implantation and to demonstrate the safety of high current density microstimulation beyond the current clinically approved limits. Several studies[66,67] have suggested the need to revisit the 30 μC/cm² limit for

microelectrodes, based on prior preclinical and clinical studies. Despite the above challenges, the advancements presented in this work could help to enhance the effectiveness, safety, and personalization of neuroelectronic therapies for neurological disorders.

## Methods

### Fabrication of rGO microelectrode array

The flexible microelectrode arrays were fabricated following the method detailed in a previous publication[18], and are thus only briefly summarized here. An aqueous solution of dispersed graphene oxide flakes was filtered to create a 2-μm-thin membrane. This membrane was then transferred onto a silicon wafer[68] that had been spin-coated with a 7.5 μm layer of polyimide (PI-2611, polyimide precursors based on BPDA/PPD, biphenyldianhydride/1,4 phenylenediamine) and evaporated with 20/200 nm layer of Ti/Au. The membrane was partially reduced hydrothermally in an autoclave. Photolithography and reactive ion etching (RIE) were used to define the rGO electrodes, while the gold tracks were defined through wet etching. A reactive ion etching step with oxigen was employed to enhance the surface oxidation level. Subsequently, an a-amino propyltriethoxysilane based adhesion promoter was applied before spin-coating the second polyimide layer to enhance the integration of the two layers. Openings of the PI were created at the electrode sites, pad areas (for the connection with the electronics), and the structures of the device using photolithography and RIE (Supplementary Fig. S1a). The completed devices were mechanically peeled off from the carrier wafer (Supplementary Fig. S1b).

The fabrication of the silicon shuttle has also been previously described[34], and it is briefly summarized here. The silicon shuttle was fabricated on a silicon-on-insulator (SOI) wafer. The structure was dry-etched using the standard Bosch process, which stops at the buried oxide layer. Finally, the shuttles were released by wet etching the buried oxide layer in 49% hydrofluoric acid.

### Electrochemical characterization

Electrochemical characterization was performed using a potentiostat (Metrohm Autolab PGSTAT128N) in a three-electrode configuration. An Ag/AgCl electrode (FlexRef, WPI) was used as the reference electrode, and a platinum wire (Alfa Aesar, 45093) as the counterelectrode. The devices were immersed in a solution prepared by dissolving one PBS tablet (Sigma-Aldrich, P4417) in 200 ml of distilled water. The final solution contains 10 mM phosphate buffer, 137 mM NaCl, and 2.7 mM KCl at pH 7.4.

Before in vitro electrochemical evaluation, the electrodes underwent an electrochemical activation process consisting of 70,000 biphasic rectangular current pulses (cathodic first with delay) of 1 ms pulse width, ramping from 5 to 20 μA at 100 Hz. Prior to in vivo implantation, a gentle reactivation was performed[25], involving 100 cycles of cyclic voltammetry from −0.9 V to 0.8 V at 50 mV/s. Electrochemical characterization in PBS involved measuring impedance, cyclic voltammetry, and current pulses to evaluate functionality, ensuring no broken or electrically shorted electrodes.

### Impedance spectroscopy

It was measured using the previously mentioned three-electrode setup by applying a 10 mV sinusoidal wave over a frequency range from 1 Hz to 100 kHz. Electrodes with impedance between 20 and 70 kΩ at 1 kHz were considered functional.

### Charge injection limit (CIL)

Biphasic current pulses were injected, and the voltage was measured against the reference. Cathodic and anodic voltage shifts were determined for each current level by visually distinguishing the voltage shift $(V_1 - V_2)$ across the electrode-PBS interface from the linear voltage produced by the resistive ohmic drop[42]. To determine the charge injection limit, voltage shift values were plotted against the current for both the cathodic and anodic parts of the pulse at increasing current levels. In Fig. 1d, the errors $(E_r)$ of the average voltage shift of the electrodes of a single array, were calculated from the standard deviation (σ) of the two voltage points $V_1 \pm \sigma_1$ and $V_2 \pm \sigma_2$, as $E_r = \sqrt{\sigma_1^2 + \sigma_2^2}$. From the linear fit of these points, we extrapolated the current values at 0.8 V and −0.9 V, representing the maximum currents safely injected within the GO water window. The CIL was then calculated by multiplying the maximum current by the pulse duration and dividing by the electrode area. Similarly, the maximum current density was calculated by dividing the maximum current by the electrode area.

### Stability

To assess the stability of the electrodes after long-term stimulation, devices were tested in PBS with the Intan stimulator in the three-electrode configuration (refer to the chronic electronic setup). 100 million biphasic current pulses (100 μs width, 50 μA in amplitude, and 1 kHz frequency) were applied through 3 electrodes. A 1 ms passive discharge, integrated into the system, was implemented between consecutive pulses to prevent charge building up at the electrode/tissue interface. Impedance and chronopotentiometry were measured with the potentiostat before and after the stimulation protocol to ensure that the electrodes did not present significant changes in their electrochemical behavior.

### Dissolution time of two adhesive polymeric coatings

Two adhesive polymers were tested in vitro by attaching polyimide-only devices to a silicon shuttle. Polyethylene glycol (PEG, 10.000 Mmol, Aldrich Sigma) was prepared by melting 5 g at 70 °C. Polyvinyl alcohol (PVA, Aldrich Sigma 9000/10.000 Mw) was prepared by stirring 20 g in 50 ml of distilled water on a hot plate at 105 °C and 400 RPM. The mixture was boiled for 3 min and then allowed to cool overnight, covered with aluminum foil.

The small tip of the dummy devices required a customized setup for alignment with the shuttle during attachment (Supplementary Fig. S3). A vacuum pump was connected through a plastic tube to a small metallic tip that held the shuttle suspended. The shuttle tip was coated with the adhesive polymer using a small brush. Under an optical microscope, the flexible lead was aligned to the shuttle by moving the stage with a micromanipulator.

The dissolution time of each adhesive polymeric coating was tested by inserting assembled devices into agarose. The agarose (Duchefa Biochemie) was prepared by stirring 0.6 g of agarose powder in 100 ml of 10 mM PBS at 400 RPM and 200 °C until the complete dissolution. The agarose was then cooled down at room temperature over several hours. The assembled devices were inserted in a beaker with agarose at 38 °C, and the tail of each device was gently pulled out every 30 s to evaluate adhesion with the shuttle. In case of detachment, the shuttle remained inside the agarose while the flexible lead was removed, determining the dissolution time for each sample.

### Measurement of insertion and buckling forces

We measured the insertion force using a load force cell (Zwick Roell, nominal force 50 N) by inserting functional devices attached to the shuttle with PVA into agar using a micro driver (Zwick Roell Z 0.5). The shuttle base was clamped to the arm controlled by the micro driver and inserted perpendicularly into agar positioned in a beaker below. The insertion speed was the same as in vivo: 100 μm/s for the first 6 mm and 5 μm/s for the next 1.5 mm. The buckling force of the shuttle was measured by contacting it against a rigid surface (the dynamometer probe station). The shuttle was clamped as previously described and moved toward the base at 100 μm/s. Upon contact with the base, the shuttle began to bend until it reached the breaking point.

## Animals and surgery

Rats were housed in ventilated cages with free access to food and water, maintained at room temperatures between 23 °C and 25 °C, and subjected to a 12-h light cycle. Animal experiments were conducted in accordance with the United Kingdom Animal (Scientific Procedures) Act 1986 and approved by the Home Office (license PPLPP9890301).

Nine Sprague Dawley rats (250–400 g) were used for acute electrophysiological recordings and another four for awake recordings. Rats received a unilateral injection of either 6-OHDA (6-hydroxy dopamine) to induce Parkinson's disease (PD model) or saline into the medial forebrain bundle (MFB) as a control. Lesioning surgery was performed under general anesthesia with isoflurane (induction at 5% in 3 l/min O2; maintained at 1.5–2% with 3 l/min O2). Buprenorphine (0.03 mg/kg) was injected subcutaneously before surgery for pain management. Body temperature was maintained at ~37 °C using a homeothermic monitoring system (Harvard Apparatus, USA). Burr holes were drilled over the STN and MFB using coordinates from the rat brain atlas (STN: AP −3.6 mm from bregma, ML −1.4 mm; MFB: AP −4.3 mm, ML −2.0 mm) using a dental drill. Histological analysis determined the placement of the device and the extent of 6-OHDA dopaminergic lesions.

## Lesioning of the nigrostriatal pathways

To induce hemi parkinsonism, 4 µl of 6-OHDA (7,5 µg/µl in 0,2% ascorbic acid dissolved in saline) (Sigma-Aldrich, USA) was injected into the MFB (DV −7.5 mm) through a 33-gauge needle connected to a 10-µl Hamilton syringe at a rate of 1 µl/min, pausing for 1 min after every 1 µl. The needle remained in place for 10 min after the injection was complete, and was then withdrawn in small steps initially, then larger steps, until fully removed. The 6-OHDA solution was freshly prepared on the morning of the lesioning surgery and kept in a cool, dark place until use. Rats were allowed to recover from surgery and returned to their cages. Behavioral signs of hemi-parkinsonism typically appeared about 2–3 weeks post-surgery.

## Behavioral tests

Fourteen or fifteen days after toxin injection, each animal was assessed using two robust behavioral tests, the Forelimb Asymmetry Test and the Apomorphine-circling test[46], to determine if hemi parkinsonism was successfully induced. Not all rats that received 6-OHDA developed motor deficits; however, 75 percent of the rats that received 6-OHDA did develop motor deficits and were included in electrophysiological studies.

## Forelimb asymmetry test

Rats with hem-parkinsonism developed motor deficits in their forelimb contralateral to the lesioned hemisphere. They became less likely to use the affected forelimb (left) when exploring their environment compared to the intact side (right). This test involved placing the rat in a transparent cylinder and counting the number of times the rat used the intact vs. affected forelimb during rearing behavior. The asymmetric ratio was calculated as $A_r = \frac{Right - Left}{Right + Left + Both}$. Properly lesioned rats preferentially used the forelimb ipsilateral to the lesion when rearing, while otherwise, rats tended to use both forelimbs equally.

## Apomorphine-circling test

To perform this test, rats were placed in a transparent cylinder and allowed to habituate for 5–10 min before receiving a subcutaneous injection of apomorphine (0.05 mg/kg, Sigma-Aldrich, USA). After administration, rats were observed for 20 min. The number of contralateral rotations (rotations in the direction opposite to the lesioned hemisphere) was counted. Rats completing more than 80 rotations within this time window were declared hemiparkinsonian[47].

## Acute in vivo experiments

**Electronic setup.** The device was connected to a 9-contact Zero Insertion Force (ZIF) Molex connector (Mouser electronics, 0.3 mm pitch), which was attached to a custom small, printed circuit board (PCB) (WURTH ELEKTRONIK). A 36-channel header connector (Digi-Key) connected the PCB to a 32-channel stimulation/recorder system (ME2100, Multichannel Systems). For the datasheet of the system: https://www.multichannelsystems.com/sites/multichannelsystems.com/files/documents/data_sheets/MCS_ME2100-System_Datasheet.pdf

**Surgery.** 2–3 weeks after the lesioning surgery, a male rat was used under isoflurane to evaluate the recording capability, while 7 female and 3 male rats were prepared for a terminal DBS experiment under urethane anesthesia (1 g/kg). Urethane was chosen as it minimally interferes with the expression of electrophysiological biomarker signals[69,70] of PD, maximizing the chances of detecting these signals. The shuttle was pre-glued with permanent glue (Loctite), and then the rGO electrodes were electrochemically activated in PBS, as previously described[25]. The shuttle was secured with double-sided tape to the PCB, which was connected to the MCS head stage, and then clamped to the arm of a micromanipulator (Scientifica, UK) (Supplementary Fig. S4b) to perform the insertion (Supplementary Fig. S4b).

Four burr holes were drilled: one over the STN (AP −3,6 mm from bregma, ML −1.4 mm) for the device, one for a Pt wire to record from the motor cortex (M1: AP: 1.0 mm, ML −2.6 mm), one for an Ag/AgCl reference wire (AP: −5.5 mm, ML: 3 mm), and one for a Pt counter electrode (AP: −5.5 mm, ML: −3 mm) (Supplementary Fig. S4a). The micromanipulator enabled controlled insertion of the device at 3 µm/s to minimize mechanical damage of the neurons. The STN was localized by looking at its characteristic spiking activity, which is quite intense compared to regions before and after that are both silent.

**Recording and stimulation.** Neuronal activity was recorded at a 40 kHz sampling rate and high-pass filtered at 1 Hz. Once the STN was reached, DBS was delivered through the electrodes inside the STN, identified by their higher spiking activity, compared to the nearby areas. Biphasic rectangular pulses with a duration of 100 µs/phase were delivered at various amplitudes (25 µA, 50 µA, 75 µA) and frequencies (10 Hz, 100 Hz, 130 Hz) for 1 min. Stimulation periods were preceded and followed by 2 minutes of recording to capture STN activity pre- and post-stimulation.

## Chronic in vivo experiments

**Electronic setup.** The device was connected to a small PCB (1.3 cm) through the same ZIF Molex connector. An 18-contact Omnetics connector (0.64 mm pitch) connected the PCB to an Intan head stage (RHS2116 16-channel, datasheet https://intantech.com/files/Intan_RHS2116_datasheet.pdf), managed by an Intan RHS 128-channel stimulation/recording controller and RHX software. This setup, integrating the amplifier directly into the chip connected to the PCB, reduced noise that would arise from cables if the amplifier were situated on the main stimulator. A 3D-printed plastic cover protected the head stage from animal scratches. Data recording and stimulation were managed with RHX Data Acquisition Software (version 3.3.1).

**Surgery.** 3–4 weeks after lesioning surgery, 4 rats underwent another surgery for permanent device implantation. Anesthesia was induced with 5% isoflurane in 70% NO2 and 30% O2 and maintained at 1–2% isoflurane. Buprenorphine analgesia (0.05 mg/kg) was administered subcutaneously at the beginning of the surgery. Videne and EMLA™ cream were applied to the scalp. Four burr holes were drilled over the STN, M1, reference, and counter electrodes using the same coordinates as previously described for the acute experiment.

Compared to the insertion strategy used in the acute experiments, which involves just one arm, for the chronic one we performed a double arm insertion (Supplementary Fig. S4a). The PCB was connected to the Intan RHS head stage and secured to the first stereotactic arm. The shuttle was attached with double-sided tape to a second arm controlled by the micromanipulator (Supplementary Fig. S4a). The device was implanted in the STN (approximately 7.5 mm deep) at an initial speed of 100 μm/s for the first 6 mm and 5 μm/s for the following 1.5 mm. When the insertion is completed, the tail is encapsulated with Kwik seal avoiding the insertion hall. 30 mMol PBS drops were applied to dissolve the PVA on the part of the device outside the brain, while brain fluids dissolved the internal coating. After sufficient time (20 min), the shuttle was removed, leaving the flexible device in the STN (Supplementary Fig. S4a). The entire assembly was secured with dental cement attached to stainless steel bone screws anchored to the skull. STN activity was recorded before, during, and after dental cement encapsulation to ensure device functionality.

**Recording and stimulation.** Recording and stimulation sessions began 24 h after surgery to allow rats to recover. At the start of each session, rats were briefly sedated with 5% isoflurane to connect the PCB to the Intan head stage. Rats were placed in a transparent cylinder, and after regaining consciousness, DBS trials began. Data were stored at a 30 kHz sampling rate with a high-pass filter of 1 Hz. Stimulation protocols similar to those under anesthesia were delivered to compare recordings from awake and anesthetized states. Recordings were obtained at multiple time points post-implantation to assess electrode performance over time. Electrochemical impedance spectroscopy at 1 kHz in chronic settings was evaluated using the Intan built-in function.

**Histology.** At the end of the experiment, rats were deeply anesthetized and perfused transcranially with 0.1 M PBS followed by 4% paraformaldehyde in 0.1 M PBS. Brains were fixed in 4% paraformaldehyde overnight at 4 °C and then transferred to 30% sucrose (4 °C). The brains were cut into 50 μm sections using a cryostat (Leica Microsystems) and processed for staining. Tyrosine hydroxylase (TH) immunofluorescent staining was used to determine the degree of degeneration of dopaminergic neurons. Brain sections were rinsed with PBS, immersed in citric acid solution (pH 6), blocked for 1 h in 10% goat serum, and incubated overnight at 4 °C in anti-TH antibody (AB152; 1:100, Millipore) in 1% BSA solution. Sections were then incubated with anti-rabbit donkey (Alexa Fluoro 647, 1:300, Code A31573) secondary antibody in 1% BSA solution for 1 h at room temperature. Sections were covered in mounting media and imaged using a slide scanner. Electrode placement accuracy was determined by obtaining sections at the level of the STN and staining with 0.2% cresyl violet (Sigma-Aldrich Co.).

**Ethics.** Every experiment involving animals has been carried out in accordance with the United Kingdom Animal (Scientific Procedures) Act 1986 and approved by the Home Office (license PPLPP9890301).

### Electrophysiological data analysis
Electrophysiological signals recorded in vivo were processed using Python packages including Numpy 1.21.5, Neo 0.11.1, pandas, seaborn 0.11.2, Matplotlib 3.5.1, Quantities 0.13.0, Elephant 0.10.0, and the custom library PhyREC 0.6.5. Origin 2018 was used for scatter plots, box and bar charts and statistical analysis.

**Spiking activity analysis.** The signal analysis involved band-stop filtering at 50 Hz and its harmonics to remove the noise of the electronics, followed by band-pass filtering in the 200 Hz–2 kHz MUA frequency range. For awake recordings, to remove the artifacts generated by chewing and rat movements, additional noise reduction was achieved by subtracting a silent channel's recording outside the STN. It

was performed the channels reorder to match the actual spatial distribution of the array, with a mapping that depends on the recording system (Intan or MCS). Spike events were detected using a voltage threshold, with a minimum relaxing time of 1.5 ms between two consecutive events. Threshold voltage levels, ranging from −20 to −30 μV, were determined individually for each experiment, based on baseline amplitude assessment. The time of detection of each spike (spikes time) was collected, and the signal was sliced in a window of 2 ms per side around these events.

To assess the spike-to-noise ratio, it was calculated the average of the maximum absolute voltage of the spikes (in the recording period of 100 s), divided by the standard deviation of the total recording signal in the MUA frequency range after noise cleaning, $SNR = \frac{\frac{1}{n}\sum_{i=1}^{n} Max|V_i(spikes)|}{STD_{signal}}$.

In the plot of Fig. 2c, it was selected for each rat the electrode in the STN with a higher SNR value. The spike rate was determined as the ratio between the number of spike events detected over the recording time considered (100 s).

**Color-coded plots.** Color-coded plots, displaying spike rate and modulation factor, were normalized to the range [0, 1] by dividing each value by the maximum in the dataset. A scatter plot visualized the normalized data, using a colormap that encoded the intensity values.

**Root mean square (RMS).** For the RMS calculation, the signal was pre-filtered for noise-cleaning (50 Hz and harmonics), within the MUA frequency range (200–2000 Hz). Then it was computed the square root of the mean of the squared signal values.

**Interspike frequency (IF) distribution.** The interspike frequency (IF) distribution was calculated as the inverse of the interspike times (IT) between two consecutive spikes detected. To ensure comparability across recordings, the cumulated IF counts were normalized by dividing for the total number of counts in the recording period. The histograms were fitted with a kernel density probability function using the displot seaborn function with kind = 'kde' and default parameters. The resulting curve represents the density of interspike frequencies, normalized such that the integral over all possible values is equal to 1. By examining these fitted histograms, it was possible to classify the IF distribution as either unimodal (typical in control rats) or bimodal (common in PD rats) for each in vivo recording.

In bimodal distributions, it was identified for each experiment a cut-off frequency (threshold), typically centered around 100 Hz, which separated the two peaks in the histogram. By calculating the maximum values above and below this threshold, it was determined the frequency center of the two peaks for each rat. For unimodal distributions, the frequency peak corresponded to the value with the maximum counts in the histogram.

**Burst and tonic events.** For bimodal distributions, burst events were defined as sequences where interspike frequencies consecutively exceeded the cut-off frequency (denoted as $IF_a$). Each sequence of consecutive frequencies above this cut-off represented one burst count. The end of a burst was identified by the presence of at least one interspike frequency below the cut-off (denoted as $IF_b$), as slow-firing neurons do not contribute to bursts. The number of spikes per burst was calculated by counting the number of $IF_a + 1$. Frequencies below the cut-off were considered tonic spike counts (tonic events).

**Burst modulation.** The evolution of the burst ratio (post/pre) over time (Fig. 4d), was determined by first calculating the average burst rate during the post-DBS period for each 15-second interval. These values were then divided by the burst rate averaged in the total

recording period pre-DBS, to obtain the post/pre ratio. The ratios were then averaged with STD across all rats. The modulation factor was calculated using the formula $F_{burst}^{mod} = \frac{Burst_{pre} - Burst_{post}}{Burst_{pre}}$, where $Burst_{pre}$ and $Burst_{post}$ represent the average burst rates over the total 100-s recording period pre and post-DBS. This factor was calculated only for channels that exhibited bursting activity.

**Cross-correlation.** The cross-correlation between the reference electrode (number 1) in the STN and each one of the others, was calculated with the correlate function in both LFP and spike frequency range and then normalized. The resulting values at lag 0 were printed for each pair.

**Local field potential analysis.** The power spectral density (PSD) was computed using the Welch method. A minimum frequency threshold (FMin) of 0.5 Hz is set, and the specific number of points for the Fast Fourier Transform (nFFT) is determined based on the sampling rate and FMin. An overlap of 2 points between consecutive segments is applied to improve frequency resolution. For average PSD, the variability between different recordings is quantified by computing the standard deviation (shadow area) around the average PSD. This is achieved by first calculating the variance (squared differences from the average PSD divided by the number of recordings) and then taking its square root. To improve the visualization of the PSD, the frequency cuts in the plot produced by the notch filters are digitally removed

**Total PSD power.** The total power within different frequency bands of the PSD was quantified by integrating the PSD across each specified frequency band for each rat.

**LFP-to-noise ratio.** The SNR of the LFP signals was calculated by dividing the PSD of signals recorded in vivo over those recorded post-mortem for each working recording channel of the array. The average SNR across channels was then computed, along with its standard deviation, to assess the quality of the recorded signals.

**Spectrogram.** It was generated using a custom library (PhyREC) with default parameters, including a frequency range of 1–200 Hz and a resolution of 0.5 Hz and 0.5 s. The color map 'jet', was normalized in a range from $10^{-13}$ to $10^{-7}$ and it was performed a bilinear interpolation.

**Fluorescence density analysis.** To compare dopaminergic neuron density between sham and Parkinsonian samples, ImageJ was used to calculate striatal fluorescence intensity following TH staining. In coronal sections, striata were visually identified in each slice by morphological characteristics, and a region of interest (ROI) was manually drawn around the structure in each hemisphere. Mean gray values (MGV) were calculated in each striatal ROI for each hemisphere. To account for variability in antibody staining and fluorescence intensity between samples, MGV from the right hemisphere (lesioned) was normalized to those from the left hemisphere (control). Normalized MGV values were then plotted and analyzed in Prism 9 (GraphPad software). A single coronal section between +1,5 and 0 mm (anterior posterior relative to bregma) was used as a representative sample of dopaminergic neuron density in each animal.

**Statistical analysis**
For each data set, it was first performed the Normality test to evaluate if the data followed a normal distribution ($P > 0.05$) or not ($P \leq 0.05$). In the first case, we quantified the statistical significance with the paired $t$ test for groups of the same size, while Welch's $t$ test for groups of different ones. In the second case, since the data are non-parametric (they don't follow a normal distribution), we performed the Mann–Whitney U Test. Significance levels were indicated as follows: NS

(not significant, $P > 0.05$), *($P \leq 0.05$), **($P \leq 0.01$), ***($P \leq 0.001$), ****($P \leq 0.0001$). All the statistical tests are two-sided. Unless otherwise specified the bar plots are expressed with median line and STD whisker. All the tests were performed with Origin 2018.

Figure 1. h, ****p = $10^{-5}$ in the Mann–Whitney U Test. i, NS: $p = 0.14$ in the Mann–Whitney U Test.

Figure 3. a, ****$p = 0.00004$ in the Welch $T$ test. b, **$p = 0.0017$ in the Welch $T$ test. c, *$p = 0.012$ in the Welch $T$ test. f, *$p = 0.045$ in the Welch $T$ test. g, **$p = 0.01$, tonic: NS $p = 0.73$ in the Welch $T$ test. k, Delta NS $p = 0.48$, theta NS $p = 0.59$, alpha NS $p = 0.68$, beta NS $p = 0.91$, low gamma NS $p = 0.43$, high gamma NS $p = 0.11$, in the Welch $T$ test for the power PD-control.

Figure 4. c, the statistics consider the post-pre values of each category in the X axis, burst *: $p = 0.015$, Tot spike NS: $p = 0.38$, Tonic NS: $p = 0.80$, Burst spike count NS: $p = 0.41$ in the paired $T$ test. d, delta NS $p = 0.23$, theta $p = 0.27$, alpha NS $p = 0.61$, beta NS $p = 0.87$, low gamma NS $p = 0.11$, high gamma NS $p = 0.62$, in the paired $T$ test.

Supplementary Fig. S7. b, magnitude after 1 week NS $p = 0.62$, after 2 weeks NS $p = 0.98$, after 3 weeks NS $p = 0.19$, in the Welch $T$ test.

## Reporting summary
Further information on research design is available in the Nature Portfolio Reporting Summary linked to this article.

## Data availability
All the data in the main text has been deposited at https://doi.org/10.34810/data1816. Supplementary experimental data that support the figures and other findings of this study can be obtained by contacting the corresponding authors. Authors can make data available on request, agreeing on the data formats needed. Source data are provided with this paper.

## Code availability
The custom code developed for neurophysiological analysis has been deposited at https://doi.org/10.34810/data1816.

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

## Acknowledgements

This research was funded by the European Union's Horizon 2020 research and innovation program under grant agreement no. 881603 (Graphene Flagship Core 3) and the European Union's Horizon Europe research and innovation program under grant agreement 101070865 (MINIGRAPH). The ICN2 has been supported by the Severo Ochoa Centres of Excellence program [SEV-2017-0706] and is currently supported by the Severo Ochoa Centres of Excellence program, Grant CEX2021-001214-S, both funded by MCIN and MCIU/AEI/10.13039.501100011033"and by the CERCA Programme of Generalitat de Catalunya. This research is also funded by the Spanish Ministerio de Ciencia e Innovación (PID2020-113663RB-I00) founded by MCIU/AEI/10.13039/ 501100011033 and by "ERDF A way of making Europe", PLEC2022-009232 and PCI2021-122095-2A, funded by MCIU/AEI/10.13039/501100011033 and the European Union Next-GenerationEU/PRTR. N. R. acknowledges grant PRE2020-093708 founded by MCIU/AEI /10.13039/501100011033 and by "ESF Investing in your future". E. M. C. acknowledges grant FJC2021-046601-I funded by MCIU/AEI/10.13039/501100011033 and the European Union Next-GenerationEU/PRTR. This work has made use of the SpanishICTS Network MICRONANOFABS, partially supported by MICINN and the ICTS NANBIOSIS, specifically by the Micro-NanoTechnology Unit U8 of the CIBER-BBN. This research was supported by CIBER -Consorcio Centro de Investigación Biomédica en Red- (CB06/01/0049), Instituto de Salud Carlos III, Ministerio de Ciencia e Innovación. We would like to thank Prof. Alex Casson from the Department of Biomedical Engineering at the University of Manchester for loaning us the potentiostat used in the experiments. The project that gave rise to these results received the support of a fellowship from the"la Caixa" Foundation (ID 100010434). M. P acknowledges grant with fellowship code LCF/BQ/DI21/11860021.

## Author contributions

N.R. worked on the design, fabrication, and characterization of the devices, development and characterization of the insertion strategy of the implant in vitro and in vivo, analysis of the in vivo data, preparation of the figures and writing of the manuscript. A.E. performed animal lesioning and device implantation surgeries, acute and chronic DBS experiments, histological tissue analysis and contributed to manuscript writing and review. E.M.C. contributed to the preparation of samples and experiments, data analysis and manuscript preparation. X.I. contributed to fabrication, A.G. contributed to data analysis and experimental preparation. K.H. performed immunohistochemical analysis, R.G.C. contributed to the flexible depth probe technology and manuscript preparation. F.T.D. contributed to the optimization of fabrication processes to improve the long-term stability of the rGO arrays. S.F. contributed to the electrophysiology experiment. M.P. contributed to the fabrication and characterization of the devices. K.K, J.A.G and R.W conceived and supervised the project, and contributed to the manuscript writing.

## Competing interests

A.G., K.K. and J.A.G. declare that they hold interest in INBRAIN Neuroelectronics which has licensed the electrode technology used in this work. All other authors declare no competing interests.
