## [Transparent Peer Review file · Nature Communications]

Flexible graphene-based neurotechnology for high-precision deep brain mapping and neuromodulation in Parkinsonian rats

Corresponding Author: Professor Jose Garrido

Version 0:

Reviewer comments:

Reviewer #1

(Remarks to the Author)

Well-written paper on nanoporous reduced graphene oxide coating of electrodes on flexible substrates for recording and stimulation for deep brain stimulation. The study works with rats as model organisms and proves placement of intracerebral probes, spike recording and effects of electrical stimulation. It is very much appreciated that the authors focus on the advantageous properties of their work without “bashing” other approaches but comparing values on a quantitative base. This is a nice example of good scientific practice. All statements are supported by quantitative data from sound experiments. This work has impact on translational research on flexible deep brain probes for recording and stimulation to eventually develop closed-loop stimulation paradigms that alter activity on a local scale. The submitted work is highly significant since it shows that these alterations might not be seen in the LFP range even though they significantly change spike activity.

There are no major general concerns or criticisms and no methodological concerns. Only minor revisions need to be done. Some more detail information should be given and some minor flaws should be corrected before the manuscript is ready for publication.

Detail comments:

Figure 1 e: left y-axis should be “ $|Z|/\text{Ohm}$ ”; wrong text

line 198: the reviewer (with an engineering background) likes the “figure of merit” expression and learned recently that this expression as technical term. Many fundamental neuroscientists, however, do not seem to know this expression (own experience). It could be good to introduce the “figure of merit” expression here before using it.

The reference list is quite balanced. It would be even stronger with some more references on polyimide probes and their interaction with the brain in the 10 micron thickness range as well as with some more references on the (absolute and comparative) in vivo performance of Pt, IrOx and PEDOT:PSS on intracortical probes (at least during recording). There have been some publications on these topics in the last year(s) which should be included.

Experimental-electrode arrays: please add the chemical abbreviation of the polyimide type you used, e.g. PDMA-PPD (or whatever you used) for good scientific practice reasons. No need to tell the supplier name.

Experimental-animals: please specify whether male, female or both sexes of rats have been used.

Experimental-recording and stimulation: Please add details on the analog low-pass filter properties of the INTAN chip. Which cut-off frequency and which order? Prove that it meets the sampling theorem and the 30 kHz. Analog filter information would be good here, since all other signal processing is done after analog-digital-conversion. Information could be also added in supplementary material as INTAN data sheet. Which electrode was the counter electrode during in vivo impedance measurement? Which amplitude was used?

SUPPLEMENTARY MATERIAL:

Figure S9: please add the 100 μs pulse width in the figure caption

Figure S10 b, d: the boxes in the graphs which should point to the fixed parameter (75 μ A, 130 Hz) are confusing the reader. Please try out another display, e.g. put the number inside the box or write something like "amplitude: 75 μ A" and "frequency: 130 Hz", respectively.

Figure S11: 100 μ s pulse width information is missing again; lower graph in (a): text and figure caption describes green and yellow but there are three colors in the graph; please add the missing one in the caption in the graph and the text.

Reviewer #2

(Remarks to the Author)

SUMMARY

The authors present a compelling and detailed description for the use of flexible graphene-based electrodes for precise deep brain mapping and neuromodulation. The results presented here paint a clear picture of the advantages that micro electrodes of reduced graphene oxide can bring to the field of deep brain stimulation. Particularly noteworthy is the ability of the electrodes to measure localized field potentials with a high signal to noise ratio thanks to the low electrode impedance and the small electrode dimensions. The stimulation capacity of the rGO electrodes exceeds that of typical electrode materials (Pt, PtIr) and is on par with more novel approaches for stimulation electrodes (IrOx, PEDOT). A clear experimental approach is presented to demonstrate the advantages of simultaneous measurement and stimulation on an adequate rat model.

The research presented here is of great relevance in the field of neuroscience, as the materials used and the methods showed, can be leveraged in other investigations and for other approaches. The integration of a stable rGO coating at the electrodes sites is of great relevance and the demonstration of the stimulation and recording capacity of the material is promising. The data collected and presented in the manuscript is sound and supports the conclusions of the authors, with an adequate number of samples used and statistical significance. The visual presentation of the results is excellent and improves the understanding and readability of the manuscript. The methods presented are thoroughly explained and can be reproduced with the amount of detail provided.

SPECIFIC COMMENTS

* An argument as to why rGO is a better material when compared against other highly porous and electrochemically active materials such as IrOx or PEDOT is missing. The authors clearly state that the rGO electrodes are comparable to these materials when comparing impedance and charge delivery, thus a clear benefit of integrating the complex rGO coating process is needed.

* As with any stimulation electrode, these are prone to physical changes as a result of the constant electrochemical and electromechanical stress caused during stimulation. Furthermore, long periods submerged in biological fluids can lead to a delamination of the electrode materials. Have the authors tested the long-term stability of this particular material combination (Au-rGO) through accelerated ageing tests and through long-term stimulation stress tests? It would be very beneficial to the claims in the manuscript that electrodes are suitable for bidirectional use (recording + stimulation) if it can be demonstrated that the stimulation over longer periods does not negatively impact the recording capabilities.

* One of the goals of the manuscript is to demonstrate that rGO electrodes are suitable for a closed-loop DBS system, however, the use of the same electrode for recording and stimulation limits the application to responsive DBS (rDBS). Have the authors considered the implementation of dedicated recording and stimulation electrodes in order to achieve adaptive DBS (aDBS)? It would be a welcome addition to the manuscript if the authors could elaborate into the reasoning for the choice of one vs the other (aDBS vs rDBS) and the advantages that the current setup presents.

* Lastly, the methods presented for the insertion and recording of biomarkers in the rat's brain are excellent, but the authors should also consider how these methods could be translated to human applications if one of the main goals of the manuscript is to cement rGO and its implementation in microelectrodes as the next big step in DBS. I encourage the authors to add a couple of lines in the summary pointing to the simplicity or complexity that this technology will face in the transition face toward real life applications

PRESENTATION NOTES

* Figure 1 - the resolution of parts a, b and g is considerably lower than the rest of the figures. Please make sure the highest quality of images is used for those parts in order to make the whole figure be as high quality as the rest. In Fig. 1-a the lines showing the distance between electrodes seem to be arbitrarily placed, please make sure these lines show the distance from centre to centre.

Reviewer #3

(Remarks to the Author)

Scientific Questions/Clarifications:

The manuscript is generally well-structured and well-written, and its content is solid. However, there are some areas of concern that need to be addressed.

While I agree that adaptive closed-loop DBS strategies would benefit from, and likely require, bidirectional electrodes, the manuscript leaves some questions unanswered. Specifically, it is unclear how the newly identified Parkinsonian biomarkers will practically close the loop, given the combination of the focal nature of the induced effects, the critical role of the distance

between stimulating and recording electrodes, and the transient effects of the stimulation. This is not to say that I do not recognize the value of this discovery, but I would have appreciated a more in-depth discussion on the feasibility of a potential new bi-directional system based on the unique combination of tonic and burst patterns.

Considering the small size of the STN, the physical constraint that only a few electrodes can fit within the target area (in rats, and I assume in humans the array would host a higher number of electrodes?), and the importance of the distance between stimulation and recording electrodes in order to create the actual closed-loop, did the authors consider a probe design with a tailored electrode arrangement that could effectively meet the clinical requirements of a Parkinson's Disease protocol? This is not a critical requirement for the present manuscript but rather a suggestion aimed at guiding future efforts toward clinical translation.

Throughout the manuscript, the authors claim that rGO (micro)electrodes are excellent closed-loop 'enablers' due to their enhanced electrochemical surface area (e.g., oxygen-related vacancies and basal plane defects creating a nanoporous structure that boosts charge injection and reduces impedance). However, this assertion is not novel in the field of neuromodulation, especially considering the extensive literature on electroactive coatings, carbon nanotube and carbon fiber-based electrodes, rough nanoPt, and more. The authors provide data on rGO electrode impedance (10 arrays) and charge injection capacity (1 mC/cm²) at which modulation is achieved, but similar results have been reported with other materials. Does rGO outperform (or at least match) other non-flat electrodes or coatings (e.g., PEDOT, nanoPt, other low-impedance carbon materials) in terms of longevity? Can the authors comment on the effects of biofouling or stimulation-induced oxidation on the performance of these electrodes over time?

It is understood that a closed-loop system is primarily intended for long-term use. I would appreciate more information on the biostability of these devices, specifically regarding their electrical and mechanical performance over time. Given that carbon-based materials are known to oxidize and regenerate by shedding the oxidized layer and revealing a fresh surface, could the authors clarify if and how rGO is expected to exhibit similar behavior?

Even though the stability of the material was addressed in the authors' previous literature, clear, referenced statements should be included in this manuscript to provide a comprehensive picture and rationale for the readers.

Regarding the safety of brain tissue when subjected to electrical stimulation and charge densities higher than 30 $\mu\text{C}/\text{cm}^2$ (a threshold supported by literature and FDA approvals), it is known that, with microelectrodes, this value may sometimes need to be exceeded to achieve clinical benefits. The authors note that the minimum charge density at which modulation is observed is 1 mC/cm². They also mention that after three weeks, it was possible to inject 50 μA - the threshold for neurostimulation - without inducing faradaic reactions that could damage electrodes or tissue. However, factors such as current density, pulse width, pulse frequency, and duty cycle can all contribute to tissue damage, as evidenced by findings from the last few decades.

This raises the question: is it sufficient to evaluate the EIS or CV of implantable electrodes and observe no major changes to conclude that no tissue damage or response has occurred? Interestingly, it has been hypothesized (10.1088/1741-2560/13/2/021001) that, in some cases, the tissue response may actually contribute to the clinical relevance of a protocol (i.e., the damage may positively impact the study's outcome). Nevertheless, the manuscript should include clear statements (backed by literature or data) on the safety of these electrodes, considering the indicated charge densities and the proposed clinical applications.

Due to the edge-effect phenomenon of disk-shaped (micro)electrodes during stimulation, tissue damage is influenced by the distance from the stimulation source. Have you investigated whether the effects induced by your stimulation protocol and the focal nature of the response are related to how the current is distributed and the distances between cells and electrodes?

How do the probe's anchors impact the removal of the probe after weeks or months, particularly following a chronic study?

Stylistic Comments/Suggestions:

I suggest avoiding sentences and paragraphs that begin with phrases like "Figure X describes/presents...". A figure should support a scientific argument rather than serve as the core focus of a key paragraph in a high-quality manuscript. Instead, the reference to "Figure X" should appear in parentheses after a statement that is being supported by the data presented in the figure.

Always include a space between a number and its unit (e.g., 1 kHz).

I recommend against using brackets around units, such as in "t [s] or t (s)." In quantity calculus, brackets are an operator that means "unit of" rather than "dimension of"

Version 1:

Reviewer comments:

Reviewer #2

(Remarks to the Author)

The current version of the manuscript addresses all of the minor reviews that were requested. The clarity and impact of the information is greatly improved and the authors have provided adequate answers to all of my questions. The additional experiment to evaluate the long-term stability of the rGO coating is appreciated as this enhances the credibility of the future applications of these devices.

Reviewer #3

(Remarks to the Author)

Reviewer #3:

- The manuscript is generally well-structured and well-written, and its content is solid. However, there are some areas of concern that need to be addressed.

While I agree that adaptive closed-loop DBS strategies would benefit from, and likely require, bidirectional electrodes, the manuscript leaves some questions unanswered. Specifically, it is unclear how the newly identified Parkinsonian biomarkers will practically close the loop, given the combination of the focal nature of the induced effects, the critical role of the distance between stimulating and recording electrodes, and the transient effects of the stimulation. This is not to say that I do not recognize the value of this discovery, but I would have appreciated a more in-depth discussion on the feasibility of a potential new bi-directional system based on the unique combination of tonic and burst patterns.

Response: We have added the following text to the conclusions section to highlight the above and emphasize on the translational challenges for future research. "Future translation of this technology to humans will face important challenges such as the implementation of a new insertion strategy^{9,10} compatible with the clinical surgical procedures and a lead design adapted to the human brain. Additionally, it will be necessary to achieve chronic recording of spiking activity over years of implantation and to demonstrate the safety of high current density microstimulation beyond the current clinically approved limits. Several studies¹¹ have suggested the need to revisit the 30 $\mu\text{C}/\text{cm}^2$ limit for microelectrodes, based on prior preclinical and clinical studies."

My question was more focused on the 'clinical/treatment' aspect, but I appreciate how the authors have addressed the practical limitations in the manuscript with the added paragraph.

- Throughout the manuscript, the authors claim that rGO (micro)electrodes are excellent closed-loop 'enablers' due to their enhanced electrochemical surface area (e.g., oxygen-related vacancies and basal plane defects creating a nanoporous structure that boosts charge injection and reduces impedance). However, this assertion is not novel in the field of neuromodulation, especially considering the extensive literature on electroactive coatings, carbon nanotube and carbon fiberbased electrodes, rough nanoPt, and more. The authors provide data on rGO electrode impedance (10 arrays) and charge injection capacity (1 mC/cm^2) at which modulation is achieved, but similar results have been reported with other materials.

Response: We believe that rGO microelectrode technology, besides the capacity to perform at the top-end of key indicators (high-charge injection and low impedance), has an interesting combination of other properties, such as microfabrication process adaptability, flexibility and thinness, biocompatibility, wide water window, chemical inertness and low cost. We have added the following to highlight additional potential benefits of the rGO technology in the main text: "Besides its state-of-the-art electrochemical and mechanical properties, rGO is also a relatively low-cost material, can be integrated with flexible microelectronics and microfabrication processes, and it is highly biocompatible^{8, 12} and chemically inert, which makes it promising for neural interfaces."

Thank you for including the other potential benefits. However, while the material is promising, it's not necessarily 'unique,' which was the point I was trying to make in my comment.

- Does rGO outperform (or at least match) other non-flat electrodes or coatings (e.g., PEDOT, nanoPt, other low-impedance carbon materials) in terms of longevity? Can the authors comment on the effects of biofouling or stimulation-induced oxidation on the performance of these electrodes over time?

Response: The rGO electrode technology has been described thoroughly earlier⁷, including some performance benchmarking and also a more thorough literature-based comparison with electrodes made of alternative materials. It is almost impossible however, to be able to make absolute statements about 'outperformance' of one versus the other materials without running simultaneously studies within the same experimental design and limitations. This is not logistically nor feasibly possible, therefore we prefer to not make statements on comparative performance in the manuscript. In this work, we present the advantage of using high-performing electrodes to improve the recording resolution and neuromodulation efficacy of Parkinsonian biomarkers in an acute preclinical disease setting. However, we do not aim to prove chronic stability of the technology. We would also like to refer Reviewer #3 to our response to Reviewer #2 comment about chronic stability, and the proof-of-concept "chronic" experiment (see Figure S5) performed to show stability in the recording of spikes with good SNR, and stable impedance module and charge injection for 3 weeks. While we are aware that this is not statistically relevant, this result would indicate a significant biofouling effect, at least in the first weeks of implantation.

I disagree with the authors' statements regarding the 'impossible' comparison, and I find it concerning that they claim not to aim to prove the chronic stability of an implantable electrode intended for closed-loop neuromodulation.

First, while it may not be 'feasible' to include other materials in the study, there is an extensive body of literature and

numerous manuscripts outlining guidelines for thorough electrode characterization and aging. Therefore, some level of comparison is not only possible but expected. This is not to suggest that the goal should be a competition over which electrode material is superior, but it's crucial not to overlook the fact that similar results can be achieved with different approaches. Presenting the current material/product as unique (either by directly saying it or implying it) can be misleading to readers in this context.

Second, according to the FDA's definition, the term chronic refers to any implantable device that is intended to function for more than 29 days in a physiological environment. If the authors wish to claim that this material enables high-precision deep brain mapping and neuromodulation, they should make a concerted effort to assess the longevity of the electrodes presented. The fact that other reviewers raised the same concern should serve as an important prompt to address this issue in depth.

- It is understood that a closed-loop system is primarily intended for long-term use. I would appreciate more information on the biostability of these devices, specifically regarding their electrical and mechanical performance over time. Given that carbon-based materials are known to oxidize and regenerate by shedding the oxidized layer and revealing a fresh surface, could the authors clarify if and how rGO is expected to exhibit similar behavior?

Response: The 'biostability' of the rGO electrodes used in this work, but without stimulation or other electrical function, has been described in a previous publication⁷ (Figures 4k,m, 5k,m in Reference [7]). We have also performed a proof-of-concept study in this manuscript (see Figure S5), to demonstrate stable impedance module and charge injection for 3 weeks. That indicates that biofouling effects or surface oxidation processes, which would increase impedance¹³, were not taking place. Also, the newly added in vitro stimulation study (Figure 1 of this response letter and new Figure S2 in Supplementary Information) does not evidence clear oxidation process.. Normally, we design our experiments to operate the electrodes within the electrochemical potential window of the rGO, which should prevent any oxidation. We have conducted extensive in situ/operando studies of the rGO material to understand the operation of the electrodes¹³. This study does not indicate irreversible oxidation processes.

To clarify, while conducting experiments within the water window prevents irreversible chemical reactions, the adsorption of chemical species onto the electrode surface can still influence electrode behavior and performance. That is why I raised the question.

I do not have further comments on the rest of the questions.

Response letter to reviewers' comments.

We would like to thank all reviewers for their thorough and thoughtful review of our work, which has contributed to improving the manuscript.

Here below, we provide a point-by-point response to their comments and suggestions.

Reviewer #1 (Remarks to the Author):

Well-written paper on nanoporous reduced graphene oxide coating of electrodes on flexible substrates for recording and stimulation for deep brain stimulation. The study works with rats as model organisms and proves placement of intracerebral probes, spike recording and effects of electrical stimulation. It is very much appreciated that the authors focus on the advantageous properties of their work without “bashing” other approaches but comparing values on a quantitative base. This is a nice example of good scientific practice. All statements are supported by quantitative data from sound experiments. This work has impact on translational research on flexible deep brain probes for recording and stimulation to eventually develop closed-loop stimulation paradigms that alter activity on a local scale. The submitted work is highly significant since it shows that these alterations might not be seen in the LFP range even though they significantly change spike activity.

We really appreciate the positive feedback of the Reviewer on the paper and the comments from the Reviewer on our scientific practices in this quite competitive field.

There are no major general concerns or criticisms and no methodological concerns. Only minor revisions need to be done. Some more detail information should be given and some minor flaws should be corrected before the manuscript is ready for publication.

Detail comments:

** Figure 1 e: left y-axis should be “ $|Z|/Ohm$ ”; wrong text*

Figure label has been updated.

** line 198: the reviewer (with an engineering background) likes the “figure of merit” expression and learned recently that this expression as technical term. Many fundamental neuroscientists, however, do not seem to know this expression (own experience). It could be good to introduce the “figure of merit” expression here before using it.*

The sentence has been updated as follows: “The ratio between both PSDs, which can be used as a figure of merit (**performance metric**) for the signal-to-noise ratio (SNR)...”

** The reference list is quite balanced. It would be even stronger with some more references on polyimide probes and their interaction with the brain in the 10 micron thickness range as well as with*

some more references on the (absolute and comparative) in vivo performance of Pt, IrOx and PEDOT:PSS on intracortical probes (at least during recording). There have been some publications on these topics in the last year(s) which should be included.

Thanks for the suggestion, we have now included a few more published work^{1,2,3,4,5,6} to complement the previous reference list in the following sentences:

- The use of neural interfaces based on μm -thin flexible substrates has been reported to be able to mitigate foreign body response of the tissue^{4,5}.
- In the 200-2000 Hz frequency range, where MUA is typically assessed, it was obtained (Figure 2d) an average spike-to-noise ratio ($\text{SNR}_{\text{spike}}$) of 7.8 ± 2.8 ($n=9$ rats), in line with what can be achieved by high-performance electrode materials^{6,3}.
- In the proof-of-concept chronic study, the in vivo impedance remained stable over the course of three weeks (Figure S6b), suggesting no significant degradation of the electrodes or changes at the electrode/tissue interface¹.
- This current level corresponds to a charge injection limit that cannot be reached by flat metal microelectrodes and is only compatible with a few electrode materials²

** Experimental-electrode arrays: please add the chemical abbreviation of the polyimide type you used, e.g. PDMA-PPD (or whatever you used) for good scientific practice reasons. No need to tell the supplier name.*

Thanks for the suggestion, we have included the requested information in the Methods section of the paper, as follows:

Experimental: Fabrication of rGO microelectrode array: "... a 7.5 μm layer of polyimide (PI-2611, polyimide precursors based on BPDA/PPD, biphenyldianhydride/1,4 phenylenediamine) and .."

** Experimental-animals: please specify whether male, female or both sexes of rats have been used.*

We have used 7 female and 4 male rats for terminal acute experiment. This information has been added in the Methods section. Un particular, in the subsection "Acute in vivo experiment, surgery"

** Experimental-recording and stimulation: Please add details on the analog low-pass filter properties of the INTAN chip. Which cut-off frequency and which order? Prove that it meets the sampling theorem and the 30 kHz. Analog filter information would be good here, since all other signal processing is done after analog-digital-conversion. Information could be also added in supplementary material as INTAN data sheet. Which electrode was the counter electrode during in vivo impedance measurement? Which amplitude was used?*

For acute experiments (most of the data of the paper), ME2100, Multichannel Systems was used: https://www.multichannelsystems.com/sites/multichannelsystems.com/files/documents/data_sheets/MCS_ME2100-System_Datasheet.pdf. For MCS the information is not available but a bandwidth from DC to 10kHz is reported.

For chronic experiments, it was used the Intan head stage (RHS2116 16-channel, datasheet https://intantech.com/files/Intan_RHS2116_datasheet.pdf). The link to the datasheet has been added to the Methods sections, in the subsection "Electronics setup".

For Intan, it was used the default analog low-pass filter setting, resulting in a bandwidth of 1-7.5 kHz. This is within the range (up to 20 kHz) available for Intan based on a maximum sampling frequency per channel of 44.6 kSamples/s. This information can be found in the datasheet, for which the link is now provided in the Methods section.

For the in vivo impedance (Fig. S6), a Pt wire was used as a counter electrode, and an Ag/AgCl wire as the reference. The measurement was performed with the built-in impedance function of the Intan system, which is moderately accurate for impedances below the M Ω range. Details can be found in the datasheet link above under the section 'Electrode Impedance Test'. The set-up and position of the wires are now included in the text in the section, "Acute and Chronic in vivo experiments, surgery".

Supplementary Material:

* *Figure S9: please add the 100 μ s pulse width in the figure caption.*

Thank you for the suggestion, Figure legend has now been updated.

* *Figure S10 b, d: the boxes in the graphs which should point to the fixed parameter (75 μ A, 130 Hz) are confusing the reader. Please try out another display, e.g. put the number inside the box or write something like "amplitude: 75 μ A" and "frequency: 130 Hz", respectively.*

Thank you for the suggestion. We have changed the indication of the frequency and current, so now it should be clearer.

* *Figure S11: 100 μ s pulse width information is missing again; lower graph in (a): text and figure caption describes green and yellow but there are three colors in the graph; please add the missing one in the caption in the graph and the text.*

Thank you for pointing this out, we have included the missing information.

Reviewer #2 (Remarks to the Author):

The authors present a compelling and detailed description for the use of flexible graphene-based electrodes for precise deep brain mapping and neuromodulation. The results presented here paint a clear picture of the advantages that micro electrodes of reduced graphene oxide can bring to the field of deep brain stimulation. Particularly noteworthy is the ability of the electrodes to measure localized field potentials with a high signal to noise ratio thanks to the low electrode impedance and the small electrode dimensions. The stimulation capacity of the rGO electrodes exceeds that of typical electrode materials (Pt, PtIr) and is on par with more novel approaches for stimulation electrodes (IrOx, PEDOT). A clear experimental approach is presented to demonstrate the advantages of simultaneous measurement and stimulation on an adequate rat model.

The research presented here is of great relevance in the field of neuroscience, as the materials used and the methods showed, can be leveraged in other investigations and for other approaches. The integration of a stable rGO coating at the electrodes sites is of great relevance and the demonstration of the stimulation and recording capacity of the material is promising. The data collected and

presented in the manuscript is sound and supports the conclusions of the authors, with an adequate number of samples used and statistical significance. The visual presentation of the results is excellent and improves the understanding and readability of the manuscript. The methods presented are thoroughly explained and can be reproduced with the amount of detail provided.

We thank the Reviewer for the positive feedback, and for recognizing the great relevance and impact that this research can have also on other investigations and approaches.

** An argument as to why rGO is a better material when compared against other highly porous and electrochemically active materials such as IrOx or PEDOT is missing. The authors clearly state that the rGO electrodes are comparable to these materials when comparing impedance and charge delivery, thus a clear benefit of integrating the complex rGO coating process is needed.*

While it is relevant to highlight the advantages and challenges related to the use of different materials, a fair comparison is generally extremely difficult since published results are generally performed on different electrode designs (areas, shapes, contacting metal material, etc.) and with varying experimental paradigms. Moreover, in this study, we do not wish to claim, nor aim to prove the superiority of rGO compared to other electrode materials. In fact, this issue was addressed in a previous publication by the authors, which is extensively cited in this manuscript⁷. In this new work, the focus is on exploring specific application aspects of the rGO technology.

We agree with the Reviewer that it may be useful to the readership to have some information about the motivation for exploring rGO electrode technology. In fact, rGO offers several interesting properties besides the competitive values of impedance and charge injection. It is a relatively low-cost material, flexible, biocompatible⁸, with a wide electrochemical potential window, and chemically inert; all in all, this makes it a promising material for neural interfaces. This information has been added to the main text of the manuscript, in the Conclusions section.

** As with any stimulation electrode, this are prone to physical changes as a result of the constant electrochemical and electromechanical stress caused during stimulation. Furthermore, long periods submerged in biological fluids can lead to a delamination of the electrode materials. Have the authors tested the long-term stability of these particular material combination (Au-rGO) through accelerated aging tests and through long-term stimulation stress tests? It would be very beneficial to the claims in the manuscript that electrodes are suitable for bidirectional use (recording + stimulation) if it can be demonstrated that the stimulation over longer periods does not negatively impact the recording capabilities.*

We thank the Reviewer for pointing out this highly relevant aspect. We totally agree that this is a milestone that every neurotechnology design needs to achieve. We will try to address this, however, we would like to emphasize that this was not the main focus of the current manuscript. This study focuses on the capabilities to record and modulate with high-efficacy multi-unit-based biomarkers.

We have now conducted an additional study to address the issue of long-term stimulation of our devices. To this end, we performed an in vitro test with the DBS protocol used to induce neuromodulation (1 mC/cm²). We have added this proof-of-concept experiment to the Supplementary Information; and we reproduce it here below.

We have also added the following sentence to the “rGO thin-film technology and microelectrode performance” section

“The stability of the microelectrodes was assessed during continuous electrical stimulation with biphasic current pulses (100 μ s pulse width and 50 μ A current amplitude, corresponding to 1 mC/cm²); after 100 million pulses, the microelectrodes do not exhibit any significant change in their characteristics (see Figure S2), in good agreement with previous work⁷”

Figure 1. a, Impedance magnitude, and phase (for 3 electrodes) measured before and after injecting 100 million stimulation pulses (biphasic current pulses of 100 μ s in pulse width and 50 μ A amplitude). **b,** Voltage polarization in response to a biphasic current pulse of 1 ms in pulse width and 5 μ A amplitude; the measurement is performed (with 3 electrodes) before and after the injection of 100 million pulses with the same protocol of subpanel a.

We would like to take the opportunity to stress that while the topic of electrode material stability is certainly critical, the long-term stability is an issue that does not only concern the electrode material, but the whole device fabrication strategy: adhesion between polymeric layers, use of inorganic encapsulating films, nature of the metals, etc. Thus, it is a very complex study that is beyond the scope of this manuscript. Our proof-of-concept stability test only demonstrates an electrode stability compatible with typical preclinical studies. However, to prove long-term stability in clinical conditions, a dedicated and very different study will be necessary.

** One of the goals of the manuscript is to demonstrate that rGO electrodes are suitable for a closed-loop DBS system, however, the use of the same electrode for recording and stimulation limits the application to responsive DBS (rDBS). Have the authors considered the implementation of dedicated recording and stimulation electrodes in order to achieve adaptive DBS (aDBS)? It would be a welcome addition to the manuscript if the authors could elaborate into the reasoning for the choice of one vs the other (aDBS vs rDBS) and the advantages that the current setup presents.*

We thank the reviewer for this question. We now take the opportunity to discuss how the closed-loop could be implemented, although this was not really the object of this study. A closed-loop study would require a full behavioural study, etc.

In this work we highlighted the bidirectionality of the electrodes and employ an experimental paradigm that could imply the use of responsive DBS, as the Reviewer points out. However, this does not exclude the possibility of using dedicated electrodes to record and stimulate. In a high-channel-

count array of densely distributed electrodes, it is possible to detect local changes in the biomarker using electrodes close to the stimulating one, hence enabling also adaptive DBS. To our knowledge, both operational paradigms (aDBS or rDBS) present conceptual and technical differences. Very importantly, potential advantages and disadvantages will not only be influenced by the electrode material but, instead the whole system constraints or specifications that will all play a critically important role.

We have included some further explanation in the Conclusions to highlight this important research line:

“This can open up the use of this technology to investigate the mechanisms around DBS and to implement closed-loop therapies where stimulation parameters are either adaptive or responsive to the biomarker level.”

Moreover, the term adaptive in front of DBS has also been removed from the abstract and main text to avoid giving the impression that this is the preferred closed-loop strategy.

** Lastly, the methods presented for the insertion and recording of biomarkers in the rat's brain are excellent, but the authors should also consider how these methods could be translated to human applications if one of the main goals of the manuscript is to cement rGO and its implementation in microelectrodes as the next big step in DBS. I encourage the authors to add a couple of lines in the summary pointing to the simplicity or complexity that this technology will face in the transition face toward real life applications.*

We agree with the Reviewer and we have added some lines to underline challenges related to the future clinical translation of the material. A new device design featuring a high-count channel array, with slightly bigger electrodes and densely distributed, will allow covering wider brain areas while keeping the capability to record and stimulate with high resolution. Moreover, aspects of insertion and chronic stability will need to be addressed.

We would like to report that the reported technology was licensed to INBRAIN Neuroelectronics, a start-up company some of us have co-founded, who are currently conducting a first-in-human clinical study with an epicortical array using the rGO technology shown in this paper. The company is developing also a device to perform deep brain stimulation for the treatment of Parkinson's disease. Our experience is that clinical translation of technology goes beyond what a research lab can do.

To reflect that, the following text has been included in the Conclusions to highlight the challenges of clinical translation:

“Future translation of this technology to humans will face important challenges such as the implementation of a new insertion strategy^{9,10} compatible with the clinical surgical procedures and a lead design adapted to the human brain. Additionally, it will be necessary to achieve chronic recording of spiking activity over years of implantation and to demonstrate the safety of high current density microstimulation beyond the current clinically approved limits. Several studies¹¹ have suggested the need to revisit the 30 $\mu\text{C}/\text{cm}^2$ limit for microelectrodes, based on prior preclinical and clinical studies.”

PRESENTATION NOTES

** Figure 1 - the resolution of parts a,b and g is considerably lower than the rest of the figures. Please make sure the highest quality of images is used for those parts in order to make the whole figure be as high quality as the rest.*

Figures were supplied in high-quality so it might be an effect of document conversion. We will ensure that the image quality is preserved for the proof version of the manuscript.

In Fig. 1a the lines showing the distance between electrodes seem to be arbitrarily placed, please make sure this lines show the distance from centre to centre.

Thanks for highlighting this, we implemented this correction in the figure.

Reviewer #3:

The manuscript is generally well-structured and well-written, and its content is solid. However, there are some areas of concern that need to be addressed.

** While I agree that adaptive closed-loop DBS strategies would benefit from, and likely require, bidirectional electrodes, the manuscript leaves some questions unanswered. Specifically, it is unclear how the newly identified Parkinsonian biomarkers will practically close the loop, given the combination of the focal nature of the induced effects, the critical role of the distance between stimulating and recording electrodes, and the transient effects of the stimulation. This is not to say that I do not recognize the value of this discovery, but I would have appreciated a more in-depth discussion on the feasibility of a potential new bi-directional system based on the unique combination of tonic and burst patterns.*

We thank the Reviewer for the positive feedback to our work and for the opportunity to discuss more in-depth how to implement a closed-loop system based on local neuromodulatory effects. The main goal of the submitted work has been to describe a “tool” that could detect wide frequency range biomarkers and modulate them with high efficacy. Yet, the clinical translation of new biomarkers will require bespoke studies, well-beyond the scope of this manuscript.

There are however several remarks we would like to make. The observation that DBS produces long-lasting modulation effects post-treatment, which is unveiled here thanks to the recording & stimulating capabilities of our technology, could open the possibility of using intermittent stimulation, with on/off periods. This could be regulated by monitoring the bursting activity with the same electrode used for stimulation, or different ones in proximity (within hundreds of micrometers apart).

The capability of recording high-quality spiking activity is particularly important, since reliable recording of spiking- activity for extended periods in humans is far from optimal. It is also possible that the use of flexible and ultrathin leads could offer signal preservation, and hence allow long-term local spiking-based closed-loop DBS. We are aware that this would require significant changes in the surgical flow, clinical procedures, etc., making clinical adoption a challenge. Yet, we believe that the development of novel tools and technologies is the way to advance the field.

We have added the following text to the conclusions section to highlight the above and emphasize on the translational challenges for future research.

“Future translation of this technology to humans will face important challenges such as the implementation of a new insertion strategy^{9,10} compatible with the clinical surgical procedures and a lead design adapted to the human brain. Additionally, it will be necessary to achieve chronic recording of spiking activity over years of implantation and to demonstrate the safety of high current density microstimulation beyond the current clinically approved limits. Several studies¹¹ have suggested the need to revisit the 30 $\mu\text{C}/\text{cm}^2$ limit for microelectrodes, based on prior preclinical and

clinical studies.”

** Considering the small size of the STN, the physical constraint that only a few electrodes can fit within the target area (in rats, and I assume in humans the array would host a higher number of electrodes?), and the importance of the distance between stimulation and recording electrodes in order to create the actual closed-loop, did the authors consider a probe design with a tailored electrode arrangement that could effectively meet the clinical requirements of a Parkinson’s Disease protocol? This is not a critical requirement for the present manuscript but rather a suggestion aimed at guiding future efforts toward clinical translation.*

Indeed, this is an important point. The human STN is approximately 300 times larger than the rat one, hence, we envision that a high-density array of small micrometer-sized electrodes (the optimum size of an electrode for human use is yet to be determined), with a few hundred micrometer pitch, could cover the STN and neighbouring regions. The simultaneous use of multiple independent electrodes at the same time would allow the injection of the needed current level in humans, maintaining the capability to be selective in both recording and stimulation and to personalize the treatment to different tissue structures and patient needs.

A brief mention to some of the challenges has been added in the Conclusion section of the revised manuscript.

** Throughout the manuscript, the authors claim that rGO (micro)electrodes are excellent closed-loop ‘enablers’ due to their enhanced electrochemical surface area (e.g., oxygen-related vacancies and basal plane defects creating a nanoporous structure that boosts charge injection and reduces impedance). However, this assertion is not novel in the field of neuromodulation, especially considering the extensive literature on electroactive coatings, carbon nanotube and carbon fiber-based electrodes, rough nanoPt, and more. The authors provide data on rGO electrode impedance (10 arrays) and charge injection capacity (1 mC/cm²) at which modulation is achieved, but similar results have been reported with other materials.*

We believe that rGO microelectrode technology, besides the capacity to perform at the top-end of key indicators (high-charge injection and low impedance), has an interesting combination of other properties, such as microfabrication process adaptability, flexibility and thinness, biocompatibility, wide water window, chemical inertness and low cost

We have added the following to highlight additional potential benefits of the rGO technology in the main text:

“Besides its state-of-the-art electrochemical and mechanical properties, rGO is also a relatively low-cost material, can be integrated with flexible microelectronics and microfabrication processes, and it is highly biocompatible^{8, 12} and chemically inert, which makes it promising for neural interfaces.”

** Does rGO outperform (or at least match) other non-flat electrodes or coatings (e.g., PEDOT, nanoPt, other low-impedance carbon materials) in terms of longevity? Can the authors comment on the effects of biofouling or stimulation-induced oxidation on the performance of these electrodes over time?*

The rGO electrode technology has been described thoroughly earlier⁷, including some performance benchmarking and also a more thorough literature-based comparison with electrodes made of alternative materials. It is almost impossible however, to be able to make absolute statements about ‘outperformance’ of one versus the other materials without running simultaneous studies within the

same experimental design and limitations. This is not logistically nor feasibly possible, therefore we prefer to not make statements on comparative performance in the manuscript.

In this work, we present the advantage of using high-performing electrodes to improve the recording resolution and neuromodulation efficacy of Parkinsonian biomarkers in an acute preclinical disease setting. However, we do not aim to prove chronic stability of the technology. We would also like to refer Reviewer #3 to our response to Reviewer #2 comment about chronic stability, and the proof-of-concept “chronic” experiment (see Figure S5) performed to show stability in the recording of spikes with good SNR, and stable impedance module and charge injection for 3 weeks. While we are aware that this is not statistically relevant, this result would indicate a significant biofouling effect, at least in the first weeks of implantation.

** It is understood that a closed-loop system is primarily intended for long-term use. I would appreciate more information on the biostability of these devices, specifically regarding their electrical and mechanical performance over time. Given that carbon-based materials are known to oxidize and regenerate by shedding the oxidized layer and revealing a fresh surface, could the authors clarify if and how rGO is expected to exhibit similar behavior?*

The ‘biostability’ of the rGO electrodes used in this work, but without stimulation or other electrical function, has been described in a previous publication⁷ (Figures 4k,m, 5k,m in Reference [7]). We have also performed a proof-of-concept study in this manuscript (see Figure S5), to demonstrate stable impedance module and charge injection for 3 weeks. That indicates that biofouling effects or surface oxidation processes, which would increase impedance¹³, were not taking place.

Also, the newly added in vitro stimulation study (Figure 1 of this response letter and new Figure S2 in Supplementary Information) does not evidence clear oxidation process. Normally, we design our experiments to operate the electrodes within the electrochemical potential window of the rGO, which should prevent any oxidation. We have conducted extensive in situ/operando studies of the rGO material to understand the operation of the electrodes¹³. This study does not indicate irreversible oxidation processes.

** Even though the stability of the material was addressed in the authors' previous literature, clear, referenced statements should be included in this manuscript to provide a comprehensive picture and rationale for the readers.*

We thank the Reviewer for this comment, that is in line with our responses to the points raised above too. To help the reader and provide a comprehensive picture of the rGO properties we have included the following text in the revised manuscript:

“The long-term modulation capability of the rGO microelectrodes was demonstrated in a proof-of-concept experiment. After 3 weeks, the electrodes could effectively modulate high-frequency brain activity (see Figure S5e). Importantly, with a charge density of 1 mC/cm², the voltage shift at the electrode-tissue interface was still within the electrochemical safe limits of the rGO water window. This result complements the data obtained in previous work,⁷ showing the recording of auditory evoked potential in mice for 3 months and the capability to inject electrical stimulation in the sciatic nerve and induce muscle activation for 2 months.”

** Regarding the safety of brain tissue when subjected to electrical stimulation and charge densities higher than 30 $\mu\text{C}/\text{cm}^2$ (a threshold supported by literature and FDA approvals), it is known that, with microelectrodes, this value may sometimes need to be exceeded to achieve clinical benefits. The authors note that the minimum charge density at which modulation is observed is 1 mC/cm². They also*

mention that after three weeks, it was possible to inject 50 μ A - the threshold for neurostimulation - without inducing faradaic reactions that could damage electrodes or tissue. However, factors such as current density, pulse width, pulse frequency, and duty cycle can all contribute to tissue damage, as evidenced by findings from the last few decades. This raises the question: is it sufficient to evaluate the EIS or CV of implantable electrodes and observe no major changes to conclude that no tissue damage or response has occurred?

We thank the Reviewer for the opportunity to explore and discuss the potential impact of stimulation on tissue damage. We fully agree with the Reviewer that EIS and CV are not sufficient to state the absence of damage to brain tissue.

Therefore, we have replaced the following text of the manuscript:

“This means that, after 3 weeks of implantation, it is possible to inject 50 μ A, the threshold for neuromodulation, without inducing faradaic reactions that can damage both tissue and electrode (Figure S5e).”

with this:

“While this result indicates a stable electrode behaviour, further investigation is needed to understand the effect of a high current density on brain tissue.”

We have also added this sentence in the Conclusions section of the revised manuscript:

“Additionally, it will be necessary to achieve chronic recording of spiking activity over years of implantation and to demonstrate the safety of high current density microstimulation beyond the current clinically approved limits. Several studies¹¹ have suggested the need to revisit the 30 μ C/cm² limit for microelectrodes, based on prior preclinical and clinical studies.”

** Interestingly, it has been hypothesized (10.1088/1741-2560/13/2/021001) that, in some cases, the tissue response may actually contribute to the clinical relevance of a protocol (i.e., the damage may positively impact the study's outcome). Nevertheless, the manuscript should include clear statements (backed by literature or data) on the safety of these electrodes, considering the indicated charge densities and the proposed clinical applications.*

We thank the Reviewer for pointing for to this relevant review. The authors share the claim that the current clinical limitation of 30 μ C/cm² should not be directly imposed to microelectrodes without further studies.

Regarding the safety of the electrodes in the reported stimulation conditions, we always intend to operate the electrodes so that the voltage polarization stays within the electrochemical-safe rGO window. We generally measure this interfacial potential in vivo when the used equipment allows (such as the Intan system in DC-mode, which can measure electrode polarization while stimulating), or in saline solution, a less accurate case.

The recording with the electrodes of the local activity of the brain tissue also provides some feedback. For instance, in the text we write *“The observed modulation of neural activity, following stimulation, is not permanent and the burst activity is gradually recovered in the time scale of hundreds of seconds, suggesting a reversible process.”*

Regarding literature¹¹, the threshold that we report to induce neuromodulation corresponds to a charge per phase of 5 nC/ph, in the range of what is reported in other studies with microelectrodes (1-

2nC/ph). Due to this relatively small charge per phase value, our electrodes fall in the non-damaging region of the Shannon plot¹⁴ (although this model has many limitations).

We have introduced the following text in the conclusions:

“Additionally, it will be necessary to demonstrate the safety of high-current density microstimulation beyond the current clinically approved limits. Several studies¹¹ have suggested the need to revisit the 30 $\mu\text{C}/\text{cm}^2$ limit for microelectrodes, based on prior preclinical and clinical studies.”

** Due to the edge-effect phenomenon of disk-shaped (micro)electrodes during stimulation, tissue damage is influenced by the distance from the stimulation source. Have you investigated whether the effects induced by your stimulation protocol and the focal nature of the response are related to how the current is distributed and the distances between cells and electrodes?*

We thank the Reviewer for the interesting hypothesis related to the focality of the stimulation. We would like to mention that the micrometer size of the electrodes (25 μm in diameter) will likely make this assessment extremely difficult from an experimental point of view. Studies combining novel techniques such as imaging methods simultaneously with stimulation, could be useful to explore these questions. We have performed some of these experiments with rigid electrodes intended for vision restoration and observed increased recruitment while increasing charge density¹⁵ (Fig. 4g in Reference [15]).

** As above further studies will be needed to assess aspects of edge-effects and focality. Studies combining novel techniques such as imaging methods simultaneously with stimulation, could be useful to explore this questions. We have performed some of this experiments with rigid electrodes intended for vision restoration and observed increased recruitment while increasing charge density (Fig. 4g, <https://pubs.rsc.org/en/content/articlelanding/2024/nh/d4nh00282b>). More studies will be required in this line to explore aspects such as focality and further studying the mechanisms of the observed neuromodulatory effects. How do the probe's anchors impact the removal of the probe after weeks or months, particularly following a chronic study?*

It will be important to evaluate the potential tissue damage produced by the anchor structure of the design. Our experiments indicate that the force generated by the flexible polyamide flaps is in the order of 1 mN, which is enough to balance the friction force at the interface polyamide shuttle and avoid the displacement of the device. This value is one order of magnitude less than the compression and stretching force generated in the tissue at the moment of the insertion (Figure S4), which indicates that the potential damage induced by the extraction of the lead is much lower than the insertion¹⁶.

Moreover, the presence of these flaps is correlated to an insertion strategy that was optimized for the rat model, and we do not think it can be directly translated to humans. Learnings from other academic and industrial approaches might be explored when translating this technology to insert flexible leads without flaps. Having said this, we think the anchors are a good approach for rodent preclinical studies.

Stylistic Comments/Suggestions:

** I suggest avoiding sentences and paragraphs that begin with phrases like “Figure X describes/presents...”. A figure should support a scientific argument rather than serve as the core*

focus of a key paragraph in a high-quality manuscript. Instead, the reference to “Figure X” should appear in parentheses after a statement that is being supported by the data presented in the figure.

We implemented this structure in the main part of the text.

* Always include a space between a number and its unit (e.g., 1 kHz).

Thanks for highlighting this, we implemented this correction.

* I recommend against using brackets around units, such as in “t [s] or t (s).” In quantity calculus, brackets are an operator that means “unit of” rather than “dimension of”

Brackets are kept for Figure labels for consistency with our previous works and other works in this journal. We will consider not using them in future works.

References

1. Gilberto Filho, Cláudio Júnior, Bruno Spinelli, Igor Damasceno, Felipe Fiuza, E. M. All-Polymeric Electrode Based on PEDOT : PSS for In Vivo Neural Recording. *Biosensors* 1–15 (2022).
2. Atefeh Ghazavi; Stuart F. Cogan. Ultramicro-sized sputtered iridium oxide electrodes in buffered saline: behavior, stability, and the effect of the perimeter to area ratio on their electrochemical properties. *Electrochim. Acta* 1–22 (2022).
3. Velasco-bosom, S., Carnicer-lombarte, A., Barone, D. G. & Malliaras, G. Electrodeposition of PEDOT:ClO₄ on non-noble tungsten microwire for nerve and brain recordings. *Mater. Adv.* 6741–6753 (2023) doi:10.1039/d3ma00949a.
4. Capuani, S., Grattoni, A., Malgir, G., Ying, C. & Chua, X. Advanced strategies to thwart foreign body response to implantable devices. 1–22 (2022) doi:10.1002/btm2.10300.
5. Guo, Z. *et al.* A flexible neural implant with ultrathin substrate for low-invasive brain – computer interface applications. 1–12 (2022) doi:10.1038/s41378-022-00464-1.
6. Hammack, A. *et al.* Ruthenium oxide based microelectrode arrays for in vitro and in vivo neural recording and stimulation. *Acta Biomater.* (2019) doi:10.1016/j.actbio.2019.10.040.
7. Viana, D. *et al.* Nanoporous graphene-based thin-film microelectrodes for in vivo high-resolution neural recording and stimulation. *Nat. Nanotechnol.* 1–22 (2023) doi:10.1038/s41565-023-01570-5.
8. Rodríguez-Meana, B. *et al.* Engineered Graphene Material Improves the Performance of Intraneural Peripheral Nerve Electrodes. *Adv. Sci.* **2308689**, 1–19 (2024).
9. Lee, K. *et al.* Flexible, scalable, high channel count stereo-electrode for recording in the human brain. *Nat. Commun.* **15**, (2024).
10. Wang, Y. *et al.* Flexible multichannel electrodes for acute recording in nonhuman primates. *Microsystems Nanoeng.* **9**, (2023).
11. Cogan, S. F., Ludwig, K. A., Welle, C. G. & Takmakov, P. Tissue damage thresholds during therapeutic electrical stimulation. *J. Neural Eng.* **13**, 21001 (2016).
12. Andrews, J. P. M. *et al.* First-in-human controlled inhalation of thin graphene oxide nanosheets to study acute cardiorespiratory responses. *Nat. Nanotechnol.* **19**, 705–714 (2024).

13. Bernicola, M. del P. *et al.* On the Electrochemical Activation of Nanoporous Reduced Graphene Oxide Electrodes Studied by In Situ/Operando Electrochemical Techniques. *Adv. Funct. Mater.* **2408441**, (2024).
14. Schiavone, G. *et al.* Guidelines to Study and Develop Soft Electrode Systems for Neural Stimulation. *Neuron* **108**, 238–258 (2020).
15. Duvan, F. T. *et al.* Graphene-based microelectrodes with bidirectional functionality for next-generation retinal electronic interfaces. *Nanoscale Horizons* 1948–1961 (2024)
doi:10.1039/d4nh00282b.
16. Fiáth, R. *et al.* Slow insertion of silicon probes improves the quality of acute neuronal recordings. *Sci. Rep.* **9**, 1–17 (2019).

Reviewer #2 (Remarks to the Author):

The current version of the manuscript addresses all of the minor reviews that were requested. The clarity and impact of the information is greatly improved and the authors have provided adequate answers to all of my questions. The additional experiment to evaluate the long-term stability of the rGO coating is appreciated as this enhances the credibility of the future applications of these devices.

We appreciate the positive feedback.

Reviewer #3 (Remarks to the Author):

Reviewer #3:

- The manuscript is generally well-structured and well-written, and its content is solid. However, there are some areas of concern that need to be addressed.

While I agree that adaptive closed-loop DBS strategies would benefit from, and likely require, bidirectional electrodes, the manuscript leaves some questions unanswered. Specifically, it is unclear how the newly identified Parkinsonian biomarkers will practically close the loop, given the combination of the focal nature of the induced effects, the critical role of the distance between stimulating and recording electrodes, and the transient effects of the stimulation. This is not to say that I do not recognize the value of this discovery, but I would have appreciated a more in-depth discussion on the feasibility of a potential new bi-directional system based on the unique combination of tonic and burst patterns.

Response: We have added the following text to the conclusions section to highlight the above and emphasize on the translational challenges for future research. "Future translation of this technology to humans will face important challenges such as the implementation of a new insertion strategy^{9,10} compatible with the clinical surgical procedures and a lead design adapted to the human brain. Additionally, it will be necessary to achieve chronic recording of spiking activity over years of implantation and to demonstrate the safety of high current density microstimulation beyond the current clinically approved limits. Several studies¹¹ have suggested the need to revisit the 30 $\mu\text{C}/\text{cm}^2$ limit for microelectrodes, based on prior preclinical and clinical studies."

\ My question was more focused on the 'clinical/treatment' aspect, but I appreciate how the authors have addressed the practical limitations in the manuscript with the added paragraph.

- Throughout the manuscript, the authors claim that rGO (micro)electrodes are excellent closed-loop 'enablers' due to their enhanced electrochemical surface area (e.g., oxygen-related vacancies and basal plane defects creating a nanoporous structure that boosts charge injection and reduces impedance). However, this assertion is not novel in the field of neuromodulation, especially considering the extensive literature on electroactive coatings, carbon nanotube and carbon fiberbased electrodes, rough nanoPt, and more. The authors provide data on rGO electrode impedance (10 arrays) and charge injection capacity (1 mC/cm^2) at which modulation is achieved, but similar results have been reported with other

materials.

Response: We believe that rGO microelectrode technology, besides the capacity to perform at the top-end of key indicators (high-charge injection and low impedance), has an interesting combination of other properties, such as microfabrication process adaptability, flexibility and thinness, biocompatibility, wide water window, chemical inertness and low cost. We have added the following to highlight additional potential benefits of the rGO technology in the main text: “Besides its state-of-the-art electrochemical and mechanical properties, rGO is also a relatively low-cost material, can be integrated with flexible microelectronics and microfabrication processes, and it is highly biocompatible^{8, 12} and chemically inert, which makes it promising for neural interfaces.”

\ Thank you for including the other potential benefits. However, while the material is promising, it's not necessarily 'unique,' which was the point I was trying to make in my comment.

- Does rGO outperform (or at least match) other non-flat electrodes or coatings (e.g., PEDOT, nanoPt, other low-impedance carbon materials) in terms of longevity? Can the authors comment on the effects of biofouling or stimulation-induced oxidation on the performance of these electrodes over time?

Response: The rGO electrode technology has been described thoroughly earlier⁷, including some performance benchmarking and also a more thorough literature-based comparison with electrodes made of alternative materials. It is almost impossible however, to be able to make absolute statements about 'outperformance' of one versus the other materials without running simultaneously studies within the same experimental design and limitations. This is not logistically nor feasibly possible, therefore we prefer to not make statements on comparative performance in the manuscript. In this work, we present the advantage of using high-performing electrodes to improve the recording resolution and neuromodulation efficacy of Parkinsonian biomarkers in an acute preclinical disease setting. However, we do not aim to prove chronic stability of the technology. We would also like to refer Reviewer #3 to our response to Reviewer #2 comment about chronic stability, and the proof-of-concept “chronic” experiment (see Figure S5) performed to show stability in the recording of spikes with good SNR, and stable impedance module and charge injection for 3 weeks. While we are aware that this is not statistically relevant, this result would indicate a significant biofouling effect, at least in the first weeks of implantation.

\ I disagree with the authors' statements regarding the 'impossible' comparison, and I find it concerning that they claim not to aim to prove the chronic stability of an implantable electrode intended for closed-loop neuromodulation.

First, while it may not be 'feasible' to include other materials in the study, there is an extensive body of literature and numerous manuscripts outlining guidelines for thorough electrode characterization and aging. Therefore, some level of comparison is not only possible but expected. This is not to suggest that the goal should be a competition over which electrode material is superior, but it's crucial not to overlook the fact that similar results can be achieved with different approaches. Presenting the current material/product as unique (either by directly saying it or implying it) can be misleading to readers in this context. Second, according to the FDA's definition, the term chronic refers to any implantable device that is intended to function for more than 29 days in a physiological environment. If the authors wish to claim that this material enables high-precision deep brain mapping and neuromodulation, they should make a concerted effort to assess the longevity of the electrodes presented. The fact that other reviewers raised the same concern should serve as an important prompt to address this issue in depth.

- It is understood that a closed-loop system is primarily intended for long-term use. I would appreciate more information on the biostability of these devices, specifically regarding their electrical and mechanical performance over time. Given that carbon-based materials are known to oxidize and regenerate by shedding the oxidized layer and revealing a fresh surface, could the authors clarify if and how rGO is expected to exhibit similar behavior?

Response: The 'biostability' of the rGO electrodes used in this work, but without stimulation or other electrical function, has been described in a previous publication⁷ (Figures 4k,m, 5k.m in Reference [7]). We have also performed a proof-of-concept study in this manuscript (see Figure S5), to demonstrate stable impedance module and charge injection for 3 weeks. That indicates that biofouling effects or surface oxidation processes, which would increase impedance¹³, were not taking place. Also, the newly added in vitro stimulation study (Figure 1 of this response letter and new Figure S2 in Supplementary Information) does not evidence clear oxidation process.. Normally, we design our experiments to operate the electrodes within the electrochemical potential window of the rGO, which should prevent any oxidation. We have conducted extensive in situ/operando studies of the rGO material to understand the operation of the electrodes¹³. This study does not indicate irreversible oxidation processes.

\ To clarify, while conducting experiments within the water window prevents irreversible chemical reactions, the adsorption of chemical species onto the electrode surface can still influence electrode behavior and performance. That is why I raised the question. I do not have further comments on the rest of the questions.

We appreciate that you found our answers comprehensive.